# Humans decompose tasks by trading off utility and computational cost

**Carlos G. Correa**[1] *, **Mark K. Ho**[2,3], **Frederick Callaway**[2], **Nathaniel D. Daw**[1,2], **Thomas L. Griffiths**[2,3]

**1** Princeton Neuroscience Institute, Princeton University, Princeton, New Jersey, United States of America,
**2** Department of Psychology, Princeton University, Princeton, New Jersey, United States of America,
**3** Department of Computer Science, Princeton University, Princeton, New Jersey, United States of America

* cgcorrea@princeton.edu

**Data Availability Statement:** The data and code used for analysis are available on GitHub at: https://github.com/cgc/resource-rational-task-decomposition.

## Abstract

Human behavior emerges from planning over elaborate decompositions of tasks into goals, subgoals, and low-level actions. How are these decompositions created and used? Here, we propose and evaluate a normative framework for task decomposition based on the simple idea that people decompose tasks to reduce the overall cost of planning while maintaining task performance. Analyzing 11,117 distinct graph-structured planning tasks, we find that our framework justifies several existing heuristics for task decomposition and makes predictions that can be distinguished from two alternative normative accounts. We report a behavioral study of task decomposition ($N = 806$) that uses 30 randomly sampled graphs, a larger and more diverse set than that of any previous behavioral study on this topic. We find that human responses are more consistent with our framework for task decomposition than alternative normative accounts and are most consistent with a heuristic—betweenness centrality—that is justified by our approach. Taken together, our results suggest the computational cost of planning is a key principle guiding the intelligent structuring of goal-directed behavior.

## Author summary

People routinely solve complex tasks by solving simpler subtasks—that is, they use a task decomposition. For example, to accomplish the task of cooking dinner, you might start by choosing a recipe—and in order to choose a recipe, you might start by opening a cookbook. But how do people identify task decompositions? A longstanding challenge for cognitive science has been to describe, explain, and predict human task decomposition strategies in terms of more fundamental computational principles. To address this challenge, we propose a model that formalizes how specific task decomposition strategies reflect rational trade-offs between the value of a solution and the cost of planning. Our account allows us to rationalize previously identified heuristic strategies, understand existing normative proposals within a unified theoretical framework, and explain human responses in a large-scale experiment.

**Funding:** This research was supported by John Templeton Foundation grant 61454 awarded to TLG and NDD (https://www.templeton.org/), U.S. Air Force Office of Scientific Research grant FA 9550-18-1-0077 awarded to TLG (https://www.afrl. af.mil/AFOSR/), and U.S. Army Research Office grant ARO W911NF-16-1-0474 awarded to NDD (https://www.arl.army.mil/who-we-are/ directorates/aro/). The funders had no role in study design, data collection and analysis, decision to publish, or preparation of the manuscript.

**Competing interests:** The authors have declared that no competing interests exist.

## Introduction

Human thought and action are hierarchically structured: We rarely tackle everyday problems in their entirety and instead routinely decompose problems into more manageable subproblems. For example, you might break down the high-level goal of "cook dinner" into a series of intermediate subgoals such as "choose a recipe," "get the ingredients from the store," and "prepare food according to the recipe." Task decomposition—identifying subproblems and reasoning about them—lies at the heart of human general intelligence. It allows people to tractably solve problems that occur at many different timescales, ranging from everyday tasks such as cooking a meal to more ambitious projects such as completing a Ph.D.

At least two questions arise in the context of human task decomposition. First, how do people use decompositions? Second, how do people decompose tasks to begin with? Existing research provides answers to these two questions, but does so largely by considering each one in isolation. For example, we know that, when given hierarchical structure, people readily use it to bootstrap learning [1, 2] and to organize planning [3, 4]. Separately, studies show how hierarchical structure emerges from graph-theoretic properties of tasks (e.g., "bottleneck" states) [5], latent causal structure in the environment [6, 7], or efficient encoding of optimal behaviors [8]. These accounts provide insights into the function and mechanisms of hierarchically structured, action-guiding representations, but, again, they largely consider *the use* and *the creation* of such representations separately.

In this paper we bridge this gap, developing an integrated account of how using a decomposition interacts with the task decomposition process itself. Our proposal is organized around a deceptively simple idea: Task decompositions are learned to facilitate efficient planning (Fig 1). Based on this intuition, we develop a normative framework that specifies how an idealized agent should choose a hierarchical structure for a domain, given the need to balance task performance with the costs of planning. We quantify planning costs straightforwardly as the run-time of a planning algorithm, which means that our framework predicts that task decomposition is the result of interactions between task structure and the algorithm used to plan. Because we quantitatively examine how cognitive costs are balanced with task performance in

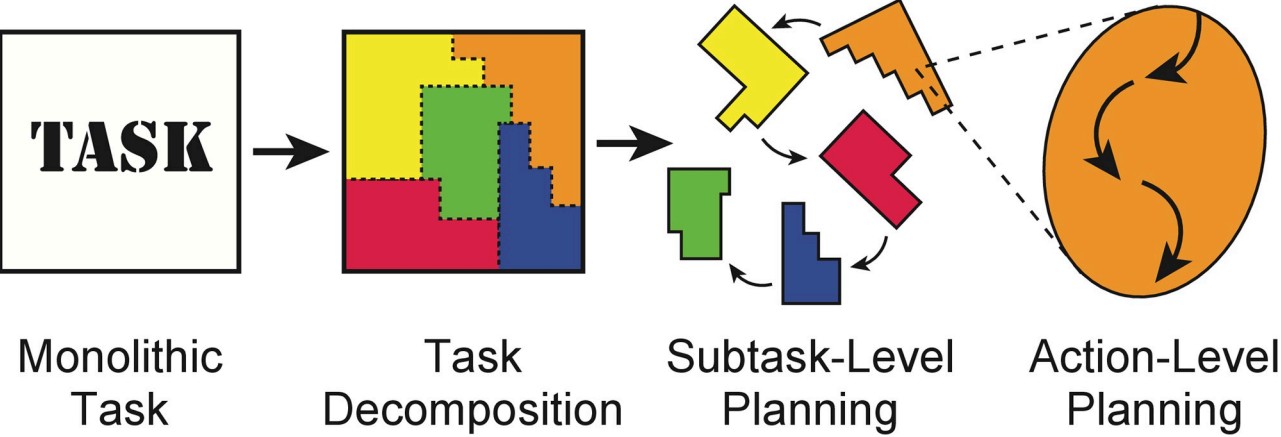

**Monolithic Task** → **Task Decomposition** → **Subtask-Level Planning** → **Action-Level Planning**

**Fig 1. Our framework for task decomposition accounts for the computational cost of planning towards subgoals—task decompositions should jointly optimize task performance and the computational cost of search.** We formalize this in three nested layers of optimization: Action-Level Planning solves for a plan to accomplish a subgoal, which has a computational cost. Subgoal-Level Planning constructs subgoal sequences that maximize reward and minimize computational cost. Task Decomposition selects subgoals based on their value in Subgoal-Level Planning. This figure was adapted from a figure published in [12] (License: CC BY 4.0).

the style of a *resource-rational analysis* [9–11], we refer to our framework as *resource-rational task decomposition*.

Although much prior research is motivated by the idea that hierarchical task decomposition has the potential to reduce planning costs [8, 13–19], our framework differs from some prominent accounts because we directly incorporate planning costs into the criteria used to choose a task decomposition. By contrast, existing normative accounts typically formulate task decomposition as *structure inference* with the goal of inferring the hierarchical structure of the environment [6, 7] or sequential behavior [8, 20], which only indirectly connect to the computations involved in planning or their efficiency. Instead, our formal framework extends research that performs task decomposition based on algorithm-specific planning costs—some algorithms previously studied are value iteration [19], random walk search [21], and random sampling of optimal behavior [8]. Generalizing beyond a fixed algorithm, our framework explicitly considers how planning efficiency shapes hierarchical representations, which we use to demonstrate how resource-rational task decompositions change with varied search algorithms. Additionally, normative accounts often have limited ability to explain human behavior without the specification of algorithmic details necessary to be efficient or psychologically plausible. Because our account focuses on efficient use of planning, it is even more critical to spell out these details. We conduct an initial exploration of these issues by examining the capacity of existing heuristics to serve as efficient algorithmic approximations to our normative account.

Using this framework, we conduct a systematic comparison of resource-rational task decomposition with four alternative formal models previously reported in the literature [7, 8, 16, 22] using 11,117 different graph-structured planning tasks. One key insight from this analysis is that our framework can justify several previously proposed *heuristics* for task decomposition based on graph-theoretic properties (e.g., those that capture the idea of "bottleneck states" [16, 22]). Our framework thus provides a normative justification for these heuristics within a broader framework of resource-rational decision-making. Critically, these connections between our framework and heuristics also demonstrate the existence of efficient approximations to our formal framework. We also show that our framework produces predictions that are distinguishable from previous normative models of task decomposition.

To empirically evaluate this framework, we report results from a pre-registered experiment ($N = 806$) that uses 30 distinct graph-structured tasks sampled from 1,676 graphs, the subset of the 11,117 graphs that are compatible with our experimental design. To our knowledge, this set of graph-structured tasks is larger and more diverse than that of any other experiment previously reported in the literature on how people decompose tasks. As such, it enables us to draw more general conclusions about task decomposition than previous studies. Across this large stimulus set, we find that our framework provides the best explanation for participant responses among normative accounts, which supports the thesis that people's hierarchical decomposition of tasks reflects a rational allocation of limited computational resources in service of effective planning and acting.

## Results

### A formal framework for task decomposition

How should an agent decompose a task? When purely optimizing behavior on a task (e.g., taking a shortest path in a graph), decomposing a task is only worthwhile in some larger context, such as making learning or computation more efficient. The computational efficiency of planning is a critical concern—as one attempts to plan into an increasingly distant future, over a

larger state space, or under conditions of greater uncertainty, computation quickly becomes intractable, a challenge termed the *curse of dimensionality* [23]. In some cases, task decomposition can ameliorate this curse by splitting a task into more manageable subtasks. The difficulty comes in choosing among task decompositions since a bad choice can make the task at hand even more difficult [1]. In this work, we formulate a framework for task decomposition where *planning costs* directly factor into people's choices—quite literally, our framework decomposes tasks into subtasks based on the run-time and utility of the plan that results from planning algorithms that solve the subtasks.

To demonstrate the role planning costs play in how people break down tasks, consider the following scenario: After leaving work for the day, you plan to go to the post office to send a letter. Since you rarely navigate directly from your workplace to the post office, you'll have to do some planning. You could determine some efficient way to get from work to the post office, but an alternative is to first get somewhere that is easy to navigate to and also along the way. Maybe the café you sometimes stop by before work—If it's easier to plan a route from the café to the post office, then you've simplified your problem by breaking it down into subtasks. This example suggests that the way people break tasks down (e.g., navigating to the café first) is a trade-off between efficiency (e.g., taking a quick route) and the cost of planning with that task decomposition (e.g., planning via the café is simpler).

We formalize our framework using three nested levels of planning and learning (Fig 1). At the lowest level is *action-level planning*, where concrete actions are chosen that solve a subtask (e.g., what direction do I walk to get to the café). The next level is *subtask-level planning*, where a sequence of subtasks is chosen (e.g., first navigating to the café and then to the post office). Finally, the highest level is *task decomposition*, where a set of subtasks that break up the environment are selected (e.g., setting the café as a possible subgoal across multiple tasks).

A central feature of our framework is the interdependence of choices made at each level: The optimal task decomposition depends on the computations occurring in the subtask-level planner, which depends on the computations that occur in the action-level planner. In particular, we are interested in how different decompositions can be evaluated as better or worse based on the cost of computing good action-level plans for a series of subtasks chosen by the subtask-level planner. In the next few sections, we discuss these different levels and how they relate to one another.

**Action-level planning.** Action-level planning computes the optimal actions that one should take to reach a subgoal. Here, we focus on deterministic, shortest-path problems. Formally, action-level planning occurs over a task $(\mathcal{S}, T, s_0, z)$ defined by a set of states, $\mathcal{S}$; an initial state, $s_0 \in \mathcal{S}$; a subgoal state, $z \in \mathcal{S}$; and valid transitions between states, $T \subseteq \mathcal{S} \times \mathcal{S}$, so that $s$ can transition to $s'$ when $(s, s') \in T$. The neighbors of $s$ are the states $s'$ that it can transition to, $\mathcal{N}(s) = \{s' \mid (s, s') \in T\}$. We refer to the structure of a task $(\mathcal{S}, T)$ as the task environment or the task graph.

Given an initial state, $s_0$, and a subgoal, $z$, action-level planning seeks to find a sequence of states that begins at $s_0$ and ends at $z$, which we denote as a plan $\pi = \langle s_0, s_1, \ldots, z \rangle$. An action-level plan is computed by a *planning algorithm*, which is a stochastic function that takes in the initial state, valid transitions, and subgoal state and takes a certain amount of time to run, $t$. Thus, we can think of a planning algorithm Alg as inducing a distribution over plans and run-times given a start and end state, $P_{\texttt{Alg}}(\pi, t \mid s, z)$. In this work, we considered four different planning algorithms [24, 25], which we summarize below.

We start with a simple algorithm that hardly seems like one—the *random walk (RW) algorithm*. The algorithm starts at the initial state $s_0$ and repeatedly transitions to a uniformly sampled neighbor of the current state until it reaches the subgoal state $z$. Because it does not keep

track of previously visited states to inform state transitions, this algorithm can revisit states many times and can result in both path lengths and run-times that are unbounded.

*Depth-first search (DFS)* augments RW by keeping track of the states along its current plan—this helps minimize repeated state visits. Because DFS does this, it will sometimes reach a dead-end where it is unable to extend the current plan, so it backtracks to an earlier state to consider an alternative choice among the other neighbors. DFS ensures that resulting plans avoid revisiting states, but still might be suboptimal and repeatedly consider the same states during the search process.

*Iterative Deepening Depth-first Search (IDDFS; [26])* consists of depth-limited DFS run to increasing depths until the goal is found; while based on DFS, IDDFS returns optimal paths because it systematically increases the depth limit. IDDFS is conceptually similar to "progressive deepening," a search strategy proposed by de Groot in seminal studies of chess players [27, 28].

*Breadth-first search (BFS)* ensures optimal paths by systematically exploring states in order of increasing distance from the start state $s_0$. The algorithm does so by considering all neighbors of the start state (which are one step away), then all of their unvisited neighbors (which are two steps away), and so on, successively repeating this process until the goal is encountered. Through this systematic process, BFS is able to guarantee optimal solutions and ensure states will only ever be considered once, making the algorithm run-time linear in the number of states.

While only noted in passing above, each of these algorithms makes subtle trade-offs between run-times, memory usage, and optimality. Focusing on BFS and IDDFS, the two optimal algorithms, we briefly examine these trade-offs. BFS visits states at most once, but requires remembering every previously visited state; by contrast, IDDFS will revisit states many times (i.e. greater run-time compared to BFS) but only has to track the current candidate plan (i.e. smaller memory use compared to BFS). In effect, IDDFS increases its run-time to avoid the cost of greater memory use. We briefly return to this point below when examining algorithm run-times. Having introduced various search algorithms, we now turn to the role algorithms play in subtask-level planning, where algorithm plans and run-times jointly influence the choice of hierarchical plan.

**Subtask-level planning.** Here, we assume a simplified model of hierarchical planning that involves only a single level above action-level planning, which we call *subtask-level planning*. Formally, subtask-level planning occurs over a set of subgoals, $\mathcal{Z} \subset \mathcal{S}$. Given a set of subgoals, subtask-level planning consists of choosing the best sequence of subgoals that accomplish a larger aim of reaching a goal state $g \in \mathcal{S}$. Each subgoal is then provided to the action-level planner, and the resulting action-level plans are combined into a complete plan to reach the goal state.

The objective of the subtask-level planner is to identify the sequence of subgoals that brings the agent to the goal state while maximizing task rewards and minimizing computational costs. Here, we focus on a domain in which the task is simply to reach the goal state in as few steps as possible. Formally, the task reward associated with executing a plan is then simply the negative number of states in that plan: $R(\pi) = -|\pi|$. The computational cost that we consider is *cumulative expected run-time*. Thus, we define a subgoal-level reward function when planning to a single subgoal $z$ from a state $s$ using a planning algorithm that induces a distribution over plans and run-times $P_{\texttt{Alg}}(\pi, t \mid s, z)$ as:

$$R_{\texttt{Alg}}(s, z) = \sum_{\pi, t} P_{\texttt{Alg}}(\pi, t \mid s, z)[R(\pi) - t]. \tag{1}$$

This formulation is analogous to other resource-rational models that jointly optimize task rewards and run-time [29, 30] but applied to the problem of task decomposition.

Eq 1 defines the rewards for planning towards a single subgoal, but subtask-planning requires chaining plans together to form a larger plan that efficiently solves the task. This sequential optimization problem can be compactly expressed as a set of recursively defined Bellman equations [23]. Formally, given a task goal $g$, a set of subgoals $\mathcal{Z}$, and an algorithm `Alg`, the optimal subtask-level planning utility for all non-goal states $s$ is then:

$$V_{\mathcal{Z}}^{g}(s) = \max_{z \in \mathcal{Z} \cup \{g\}} \left\{ R_{\mathrm{Alg}}(s, z) + V_{\mathcal{Z}}^{g}(z) \right\} \tag{2}$$

To ensure this recursive equation terminates, the utility of the goal state $g$ is $V_{\mathcal{Z}}^{g}(g) = 0$. The fixed point of Eq 2 can be used to identify the optimal subtask-level policy [31]. We permit the selection of the goal $g$ as a subgoal to ensure that it is possible for the subtask-level planner to solve the task.

**Task decomposition.** Having defined action-level planning and subtask-level planning over subgoals, we can now turn to our original motivating question: How should people decompose tasks? In this context, this reduces to the problem of selecting the best set of subgoals $\mathcal{Z}$ to plan over. Importantly, we assume that people rely on a common set of subgoals for all the different possible tasks that they might have to accomplish in a given environment. Thus, the value of a task decomposition, $\mathcal{Z}$, is given by the value of the subtask-level plans averaged over the task distribution of an environment, $p(s_0, g)$. That is,

$$V(\mathcal{Z}) = \sum_{s_0, g} p(s_0, g) V_{\mathcal{Z}}^{g}(s_0). \tag{3}$$

The optimal set of subgoals $\mathcal{Z}^{*}$ for planning maximize this value, so $\mathcal{Z}^{*} = \mathrm{argmax}_{\mathcal{Z}} V(\mathcal{Z})$.

To summarize, the value of a task decomposition (Eq 3) depends on how a subtask-level planner plans over the decomposed task (Eq 2), which is shaped by the resulting plans and run-time of action-level planning (Eq 1). This model thus captures how several key factors shape task decomposition: the structure of the environment, the distribution of tasks given by an environment, and the algorithm used to plan at the action level.

To provide an intuition for our framework, we explore its predictions in a simple task in Fig 2. The environment is a grid with a single task that requires navigating from the green state to the orange state. Each column in the figure corresponds to a different search algorithm, showing how search costs change without subgoals (Top) and with a subgoal (Bottom; subgoal in blue). While the task seems like it would be extraordinarily trivial to a person—like walking from one side of a room to another—a critical attribute of these search algorithms is they have an entirely unstructured representation of the environment, giving them only very local visibility at a state. A more analogous task for a person might be navigating in a place with low visibility, such as a forest or a city in a blackout. Even in this simple task, some search algorithms (BFS and IDDFS) can be used more efficiently when the problem is split at its midpoint (Fig 2d). The random walk is a notable counterexample, where using a subgoal results in less efficient search. This result may initially seem puzzling, but occurs because a random walk is likely to get to the goal without passing through the subgoal. The gap in run-time between IDDFS and BFS might make IDDFS seem inefficient—however, as noted in the algorithm descriptions above, IDDFS makes a trade-off of increased run-time in order to decrease memory usage. While outside the scope of the current manuscript, our formulation can be extended to incorporate other resource costs like memory usage in order to study how they influence task decomposition. This example clearly demonstrates a few characteristics of our framework— that the choice of hierarchy critically depends on the algorithm used for search, and that

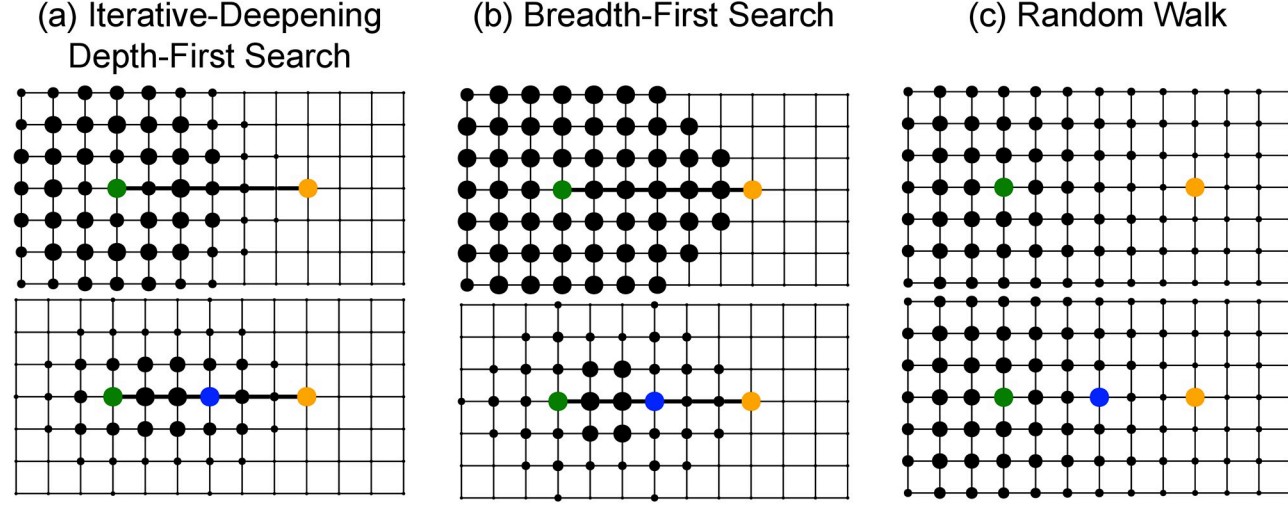

(a) Iterative-Deepening Depth-First Search   (b) Breadth-First Search   (c) Random Walk

(d)

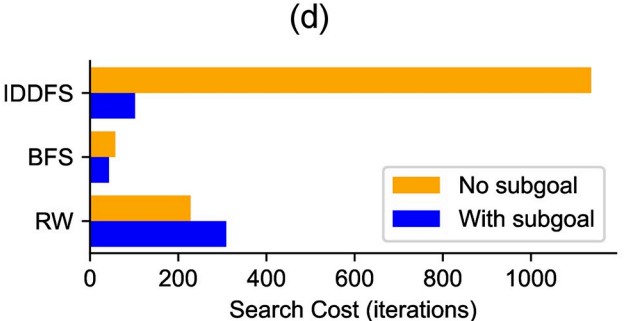

**Fig 2. Choosing task decompositions that make planning more efficient.** (a-c) State-specific search costs of the algorithms. The depicted task requires navigating on a grid from the start state (green) to the goal state (orange) with the fewest steps. Each column corresponds to a different algorithm and demonstrates two scenarios—*Top*: Search cost without subgoals, *Bottom*: Search cost when using the path midpoint as a subgoal (blue). We define the search cost as the number of iterations required for the search algorithm to find a solution. Larger states were considered more often during the search algorithm, resulting in greater search costs. (d) Plot of search cost for Iterative-Deepening Depth-First Search (IDDFS), Breadth-First Search (BFS), and a Random Walk (RW) in the task depicted in (a-c), with and without subgoals. Use of subgoals results in decreased search cost for BFS and IDDFS, but not for RW.

hierarchy can have a normative benefit (since it reduces computational costs) even in the absence of learning or generalization.

The formal presentation of our framework considers subgoal choice with intentionally restricted algorithms: brute-force search methods that exclude problem-specific heuristics to accelerate planning. However, the examples in this section (navigating to the post office via the café, navigating in a place with low visibility) likely rely on algorithms that incorporate heuristics, particularly related to spatial navigation. While outside the scope of this manuscript, our framework can flexibly incorporate any search algorithm that can define an algorithmic cost, including those that make use of heuristics. For example, in a previous theoretical study, we applied an early version of our framework to task decomposition in the Tower of Hanoi by using A* Search [25] with an edit distance heuristic [12]. Our framework also considers a constrained set of task decompositions that consist of individual subgoals. In comparison, the influential *options framework* [15] defines a more general set of hierarchical task decompositions, in which, for instance, subtasks can be defined by the subgoal of reaching any one out of

a set of states and subtask completion can be non-deterministic. However, finding such subtasks remains a challenging problem in machine learning, so by focusing on the simpler problem of selecting a single subgoal we are able to make a significant amount of progress in understanding a key component of how humans plan. While not yet fully explored, our formalism can be extended to encompass broader types of hierarchy (and also varied search algorithms). For instance, another theoretical study adapted our framework to support more varied kinds of hierarchical structure by incorporating abstract spatial subgoals in a block construction task [32].

## Comparing accounts of task decomposition

A number of existing theories have been proposed for how people decompose tasks. These accounts can be divided into two broad categories: *heuristics* for decomposition based on graph-theoretic properties of tasks and *normative* accounts based on the functional role of a decomposition. Our account is normative, so comparison with alternative normative accounts highlights the unique functional consequences of our framework. By comparing our framework to heuristics, we can characterize their predictions relative to normative theories as well as rationalize their use as heuristics for task decomposition. All models are listed and briefly described in Table 1.

To begin with, one way to compare accounts is to relate them formally. We do so by relating resource-rational task decomposition with a random walk (RRTD-RW) to QCut [16, 33] and Degree Centrality in S1 Appendix. We prove a relationship between RRTD-RW and QCut that connects the two methods through spectral analysis and examine how the relationship between RRTD-RW and Degree Centrality varies based on spectral graph properties.

Another method we can use to compare theories is to compute their subgoal predictions on a fixed set of environments and compare them—qualitatively or quantitatively. This approach has been used in existing studies, but nearly always through qualitative comparison using a small number of hand-picked environments. For example, several published models perform qualitative comparisons to the graphs studied in Solway et al. (2014) [8]. These environments contain states that most models agree should be subgoals [7, 12, 21, 34]—these states are

**Table 1. Descriptions of Normative Algorithms and Heuristics.**

| Normative Algorithm | Description |
|---|---|
| RRTD-IDDFS | Resource-Rational Task Decomposition (RRTD) using Iterative-Deepening Depth-First Search (IDDFS) as a search algorithm |
| RRTD-BFS | RRTD using Breadth-First Search (BFS) as a search algorithm |
| RRTD-RW | RRTD using a Random Walk (RW) as a search algorithm |
| Solway et al. (2014) [8] | Identifies partitions of the task into subtasks that minimize the description length of optimal solutions, given that subtask solutions are reused across tasks. |
| Tomov et al. (2020) [7] | Performs inference over partitions of the task graph into regions based on a prior over hierarchical graphs. Incorporates a preference for tasks to start and end in the same region, and for states in the same region to have similar rewards. |
| Heuristic | Description |
| QCut [16, 33] | Partitions the task graph through spectral decomposition of the graph. |
| Degree Centrality | Chooses subgoals based on Degree Centrality, which is the number of transitions into or out of a state $s$. For tasks where all state transitions are reversible, Degree Centrality is the number of neighbors $|\mathcal{N}(s)|$. |
| Betweenness Centrality [22] | Chooses subgoals based on Betweenness Centrality, which is how often a state $s$ appears on shortest paths, averaged over all possible start and goal states. Takes into account cases with multiple shortest paths. |

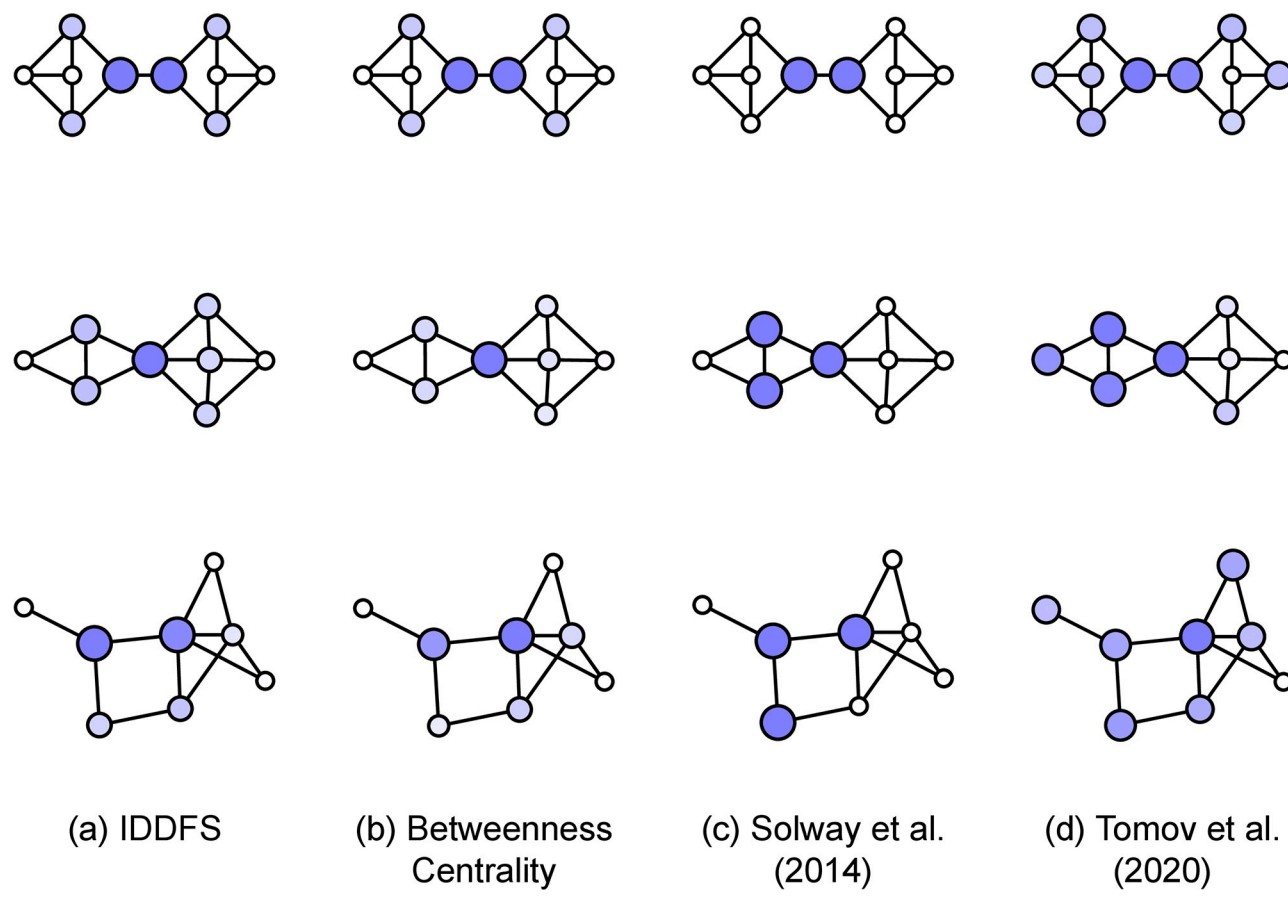

**Fig 3. Comparing predictions of the (a) RRTD-IDDFS, (b) Betweenness Centrality, (c) Solway et al. (2014) [8], and (d) Tomov et al. (2020) [7] models.** State color and size is proportional to model prediction when using the state as a subgoal. (Top) The 10-node, regular graph from Solway et al. (2014) [8]. (Middle, Bottom) Two eight-node graphs selected from the 11,117 included in our analysis.

typically one or few that connect otherwise disconnected parts of the environment, making them "bottleneck states." Environments with these kinds of bottleneck states robustly elicit hierarchically-structured behavior in experiments, but make it difficult to distinguish among theoretical accounts because they are in strong agreement (see top row of Fig 3).

To perform a large-scale and unbiased comparison of these algorithms, we chose from a structurally rich set of environments: the set of all possible 11,117 simple, undirected, eight-node, connected graphs. We compare subgoal choice for several heuristic theories, several variants of our framework, and variants of the normative accounts proposed by Solway et al. (2014) [8] and Tomov et al. (2020) [7] in Fig 4 (the theories are described in Table 1). Each cell of Fig 4 is the correlation between the subgoal predictions of a pair of theories, averaged across all environments. For simplicity, we assume the task distribution is a uniform distribution over pairs of distinct start and goal states. For the RRTD-based models, the model prediction for a state is the corresponding value of task decomposition when the state is the only possible subgoal.

We find a few notable clusters of theories—one demonstrates that RRTD-IDDFS is well-correlated with subgoal sampling based on Betweenness Centrality—this suggests the potential of a formal connection between the two algorithms, though the authors are not aware of an existing proof connecting them. A second prominent cluster shows RRTD-RW and Degree

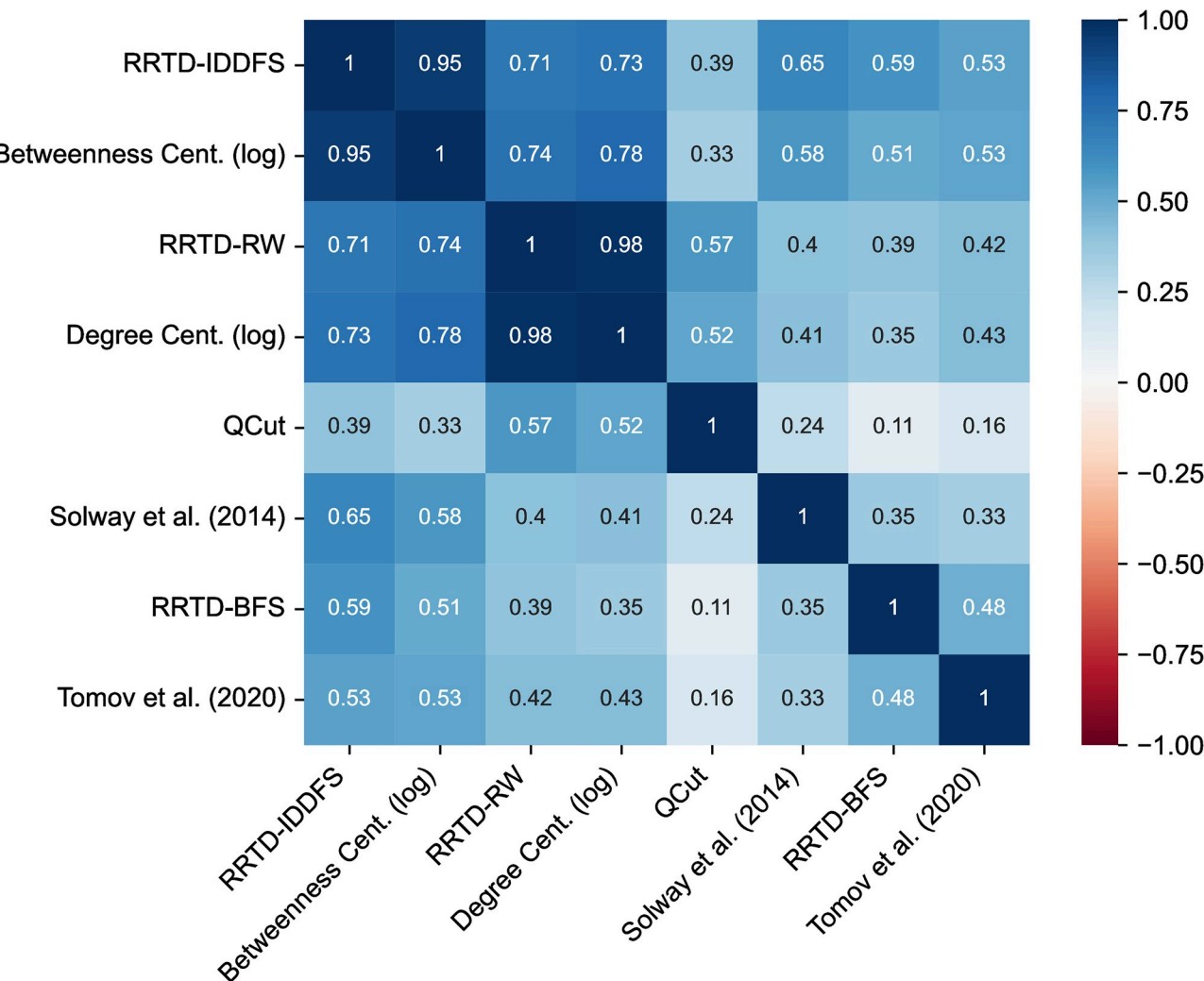

**Fig 4. Correlation matrix comparing model predictions.** For each graph, correlations between two models are computed on the per-state subgoal values, then averaged across the 11,117 simple, connected, undirected, eight-node graphs. We discard correlations when either of the two models predicts a uniform distribution over subgoals because the correlation is not defined in those instances. References: Solway et al. (2014) [8], Tomov et al. (2020) [7].

Centrality are highly correlated and that both are moderately correlated with QCut, consistent with our formal analysis. This relationship between RRTD-RW and Degree Centrality is qualitatively consistent with a published result that relates Degree Centrality to the task decomposition that minimizes a search cost related to RW [21]. The remaining algorithms—RRTD-BFS, Solway et al. (2014) [8], and Tomov et al. (2020) [7]—are singleton clusters, suggesting qualitative differences from the other algorithms.

To better understand these large-scale quantitative patterns, we qualitatively examine some of the normative algorithms and one heuristic. We focus on the subgoal predictions for three graphs, shown in Fig 3. In the top row is a 10-node, regular graph that has been previously studied [7, 8]. In the middle row is a similar graph with critical differences: the graph is asymmetric about the graph bottleneck, and the bottleneck of the graph now corresponds to a single state instead of two connected states. In the bottom row is a graph notably distinct from graphs

typically studied because it lacks an obvious hierarchical structure. All algorithms make similar predictions for the graph in the top row—an example of the difficulty in using typically studied graphs to distinguish among algorithms. Now, we look at the algorithms in more detail.

We first examine RRTD-IDDFS (Fig fig:graph-examples-smalla) and Betweenness Centrality (Fig 3b), noting their strong agreement—this is consistent with the large-scale correlation analysis in the previous section. These two algorithms prefer the same subgoals in both graphs. At middle, they prefer the bottleneck state. At bottom, they prefer states that are close to many states—in particular, the two most-preferred states can reach any other state in at most two steps.

Now, we turn to the other normative accounts: Solway et al. (2014) [8] (Fig 3c) and Tomov et al. (2020) [7] (Fig 3d). Both rely on partition-based representations of hierarchical structure where states are partitioned into different groups. Mapping from partitions onto subgoal choices requires a step of translation. In particular, when considering a path that crosses from one group into another there are two natural subgoals that correspond to the boundary between the groups: either the last state in the first group or the first state in the second group. In the context of a task distribution, there are many possible ways to map partitions onto subgoal choices, without clear consensus between the two partition-based accounts that we consider. While these analysis choices have little impact on symmetric graphs (e.g., top row of Fig 3), they are important for asymmetric graphs like the one in the middle row, which has a bottleneck state instead of a bottleneck edge. For simplicity, our implementation of Solway et al. (2014) [8] uses the optimal hierarchy, placing uniform weight over all states at the boundaries between groups of the partition, as can be seen in Fig 3c.

The algorithm from Tomov et al. (2020) [7] introduces other subtleties. It poses task decomposition as inference of hierarchical structure, with two main criteria: 1) that there are neither too few nor too many groups (accomplished via a Chinese Restaurant Process) and 2) that connections within groups are dense while connections between groups are sparse. The latter leads to issues when connection counts do not reflect hierarchical structure, as shown in Fig 3d. At middle, the algorithm prefers partitions that minimize the number of cross-group connections, even when the bottleneck state is not on the boundary between groups. At bottom, the lack of hierarchical structure that can be detected by edge counts leads the algorithm to make diffuse predictions among many possible subgoals.

To close this section, we briefly discuss how our resource-rational account might be plausibly implemented. We outline two broad approaches: people might directly search for task decompositions that maximize Eq 3, or they might attempt to approximate the objective through tractable heuristics. First, finding the optimal task decomposition in a brute-force manner is more computationally expensive than simply solving the task. One alternative is to *learn* the value of task decompositions, relying on the shared structure between tasks and subgoals to ensure learning efficiency—for example, in the domain of strategy selection, one study uses shared structure to ensure efficient estimation which is incorporated by decision-theoretic methods to deal with the uncertainty in these estimates [35]. The second approach might approximately optimize the objective by using a more tractable heuristic—the results in this section suggest two examples, where Betweenness Centrality can approximate RRTD-IDDFS and Degree Centrality can approximate RRTD-RW. While Degree Centrality is straightforward to compute, Betweenness Centrality is still computationally costly because it requires finding optimal paths for all tasks. Importantly, Betweenness Centrality has a probabilistic formulation, so it can be estimated with analytic error bounds [36]. In this formulation, states that are more central appear more often in paths sampled from an appropriate distribution (i.e. sample a task, then sample an optimal path uniformly at random). This suggests a trivial memory-based strategy that tracks the occupancy of states visited along paths—when the

paths are appropriately sampled, the expected occupancy should be related to Betweenness Centrality. Another approach is to approximate Betweenness Centrality, like in one planning-specific method that analyzes small regions of the environment separately, then pools this information to choose subgoals [22].

The large-scale comparison of subgoal predictions in Fig 4 demonstrates connections between existing heuristics and our framework for subgoal choice based on search efficiency. These connections suggest a rationale for the efficacy of these heuristics, which may stand in as tractable approximations of our resource-rational framework. Our qualitative comparison in Fig 3 highlights some of the differing predictions among the accounts. But how do these different accounts relate to how people decompose tasks? We turn to this question in the next section.

## An empirical test of the framework

To measure people's task decompositions, we ask research participants to report their subgoal use after experience navigating in an environment. A previously published experiment by Solway et al. (2014) [8] tested task decomposition using three graph navigation tasks. We developed a similar paradigm but used a set of 30 environments sampled randomly from 1,676 graphs, a subset of the 11,117 used in our large-scale model comparison above. The criteria for this subset were selected to ensure compatibility with the experiment and are detailed in the methods. This set of environments is larger and more diverse than those used in previous studies [8, 37] which allows us to draw broader and more generalizable inferences about the task decomposition process.

Inspired by prior studies [8], we conducted an experiment with two phases: participants were first familiarized with an environment by performing a series of *navigation trials* (Fig 5a), then answered a series of questions about their subgoal choices (Fig 5c and 5d).

Participants gained exposure to the environment by performing 30 navigation trials requiring navigation to a goal state from some initial state. These *long trials* were randomly selected while ensuring the initial and goal states were not directly connected (Fig 5a). Participants moved from the current state to a neighboring state using numeric keys. The trial ended when the goal state was reached. In simulations and pilot studies, states with high visit rates coincided with the predictions of RRTD-IDDFS and Betweenness Centrality. This made it difficult to dissociate model predictions from an alternative memory-based strategy where frequently visited states are selected as subgoals. As noted in the previous section, this memory-based strategy is related to sampling-based estimation of Betweenness Centrality. To address this confound, we modified the experimental task distribution so that long trials were interleaved with *filler trials* requiring navigation to a state directly connected to the start state. These filler trials were adaptively selected to increase visits to states besides the most frequently visited one; in pilot studies and simulations, this was sufficient to dissociate visit rate and model predictions.

A critical methodological difficulty is visually representing the environment in a way that enables rapid learning but does not introduce confounds. To prevent participants from relying on heuristics such as the Euclidean distance between states, states were assigned random locations in a circular layout. The absence of useful heuristics ensures the problems participants face are more comparable to that solved by brute-force search algorithms, which also lack a heuristic. To encourage model-based reasoning instead of visual search, connections between states were only shown for the current state. So that participants could still easily learn the connections, participants were periodically shown the graph with all connections between trials (Fig 5b).

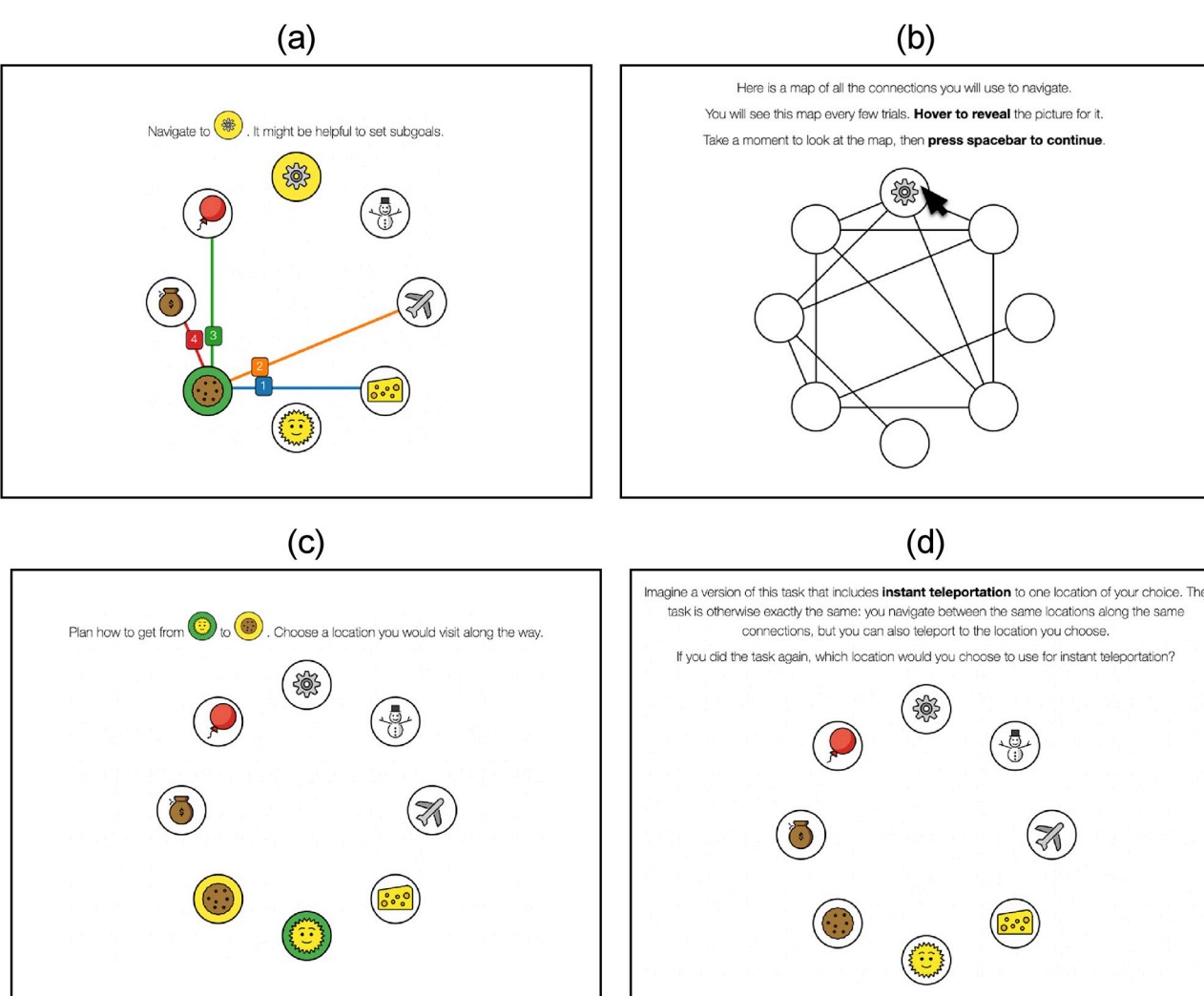

**Fig 5. Screenshots from the experimental interface.** The depicted graph is the same as the top left graph of Fig 10. (a) An example *navigation trial*. The current state has a green background and the goal state has a yellow background. Only the edges connected to the current state are shown. (b) The interface used to show all graph edges between navigation trials. There is no indication of past or future trials on this screen. State icons are only shown when the cursor is placed on them. (c) An example *implicit subgoal probe*. (d) The final post-task assessment with the *teleportation question*. All icons were designed by OpenMoji and are reproduced here with permission.

To query participant subgoal choice, we used both direct and indirect probes to comprehensively and reliably measure subgoal choice, including novel as well as previously studied prompts [8]. In the context of 10 navigation trials, we first prompted participants "Plan how to get from A to B. Choose a location you would visit along the way," the *implicit subgoal probe* (Fig 5c). Then, in the context of the same trials after shuffling, we asked participants "When navigating from A to B, what location would you set as a subgoal? (If none, click on the goal)," the *explicit subgoal probe*. In order to ensure familiarity with the concept of a subgoal, participants were introduced to the concept of a "subgoal" in the context of a cross-country road trip during the experiment tutorial. In a final post-task assessment, we asked participants "If you did the task again, which location would you choose to use for instant teleportation?", the *teleportation question* (Fig 5d). We asked this question outside the context of any particular navigation trial.

### Experiment results

We recruited English-speaking participants in the United States on the Prolific recruiting platform, prescreening to exclude participants of previous experimental pilots and those with approval ratings below 95%. Of the 952 participants that completed the experiment, 806 (85%) satisfied the pre-registered exclusion criteria requiring efficient performance on the navigation trials. If a participant took 75% more actions than the optimal path (averaged across the last half of long trials), their data was excluded. The number of participants per graph varied after exclusion criteria were applied, without significant differences per graph (before exclusion: range 27–34, after exclusion: range 21–30, two-factor $\chi^2$ test comparing included to excluded, $p = .985$). Participants took an average of 17.17 minutes ($SD = 8.02$) to complete the experiment.

Even though the experimental interface obfuscated task structure by showing the task states in a random circular layout, participants became more effective from the first to the second half of training: long trials were solved more quickly (from 10.30s ($SD = 29.74$) to 7.60s ($SD = 11.19$)), with more efficient solutions (from 36% to 20% more actions than the optimal path; completely optimal solutions increased from 70% to 79%; solutions that included a repeated state decreased from 14% to 9%), and with decreased use of the map (on-screen duration decreased from 9.01s ($SD = 20.97$) to 2.98s ($SD = 12.66$); number of hovered states decreased from 5.43 ($SD = 8.70$) to 1.38 $SD = 3.99$; duration of state hovering decreased from 2.52s ($SD = 7.23$) to 0.60s ($SD = 9.04$)). In order to rule out confounding effects due to differences in the complexity and structure of tasks that participants solved, we related participant behavior on the probes to measures associated with each graph and found no significant relationships in S1 Appendix.

Our findings are organized into two sections: First, an analysis of the subgoal probes, demonstrating their internal consistency and relationship to behavior. Then, model-based prediction of subgoal probe choice, as well as a subset of choice behavior.

**Subgoal probes are internally consistent and predict behavior.**   A crucial methodological concern is the validity of the probes for subgoal choice—in existing studies, various types of probes have been used, but not compared systematically. Choice on the explicit and implicit subgoal probes had high within-probe consistency across participants while choice on the teleportation question had low within-probe consistency across participants, based on the average correlation between per-participant choice rates and per-graph choice rates (Explicit Probe $r = 0.71$, Implicit Probe $r = 0.63$, Teleportation Question $r = 0.37$). We also evaluated consistency between probes by comparing the per-graph, per-state choice rates. The Explicit and Implicit Probes were well-correlated ($r = 0.98$, $p < .001$), though the relationship between the Teleportation Question and the remaining probes was weaker (Teleportation Question and Explicit Probe: $r = 0.58$, $p < .001$, Teleportation Question and Implicit Probe: $r = 0.58$, $p < .001$). While the Teleportation Question exhibits relatively low self-consistency and cross-probe consistency, it is difficult to compare to the consistency of the other probes since both the Explicit and Implicit probes were sampled for 10 different tasks per participant, while the Teleportation Question was only sampled once per participant.

Beyond simply assessing consistency, it is also crucial to link the probes to participant behavior during navigation, ensuring there is a link between decision-making and our indirect assessment via probes. On the Explicit Probe trials, participants were given the option of choosing the goal instead of a subgoal. On average, participants who chose a subgoal more frequently took shorter paths in the navigation trials ($r = -0.29$, $p < .001$; Fig 6), which suggests that use of subgoals promotes efficiency.

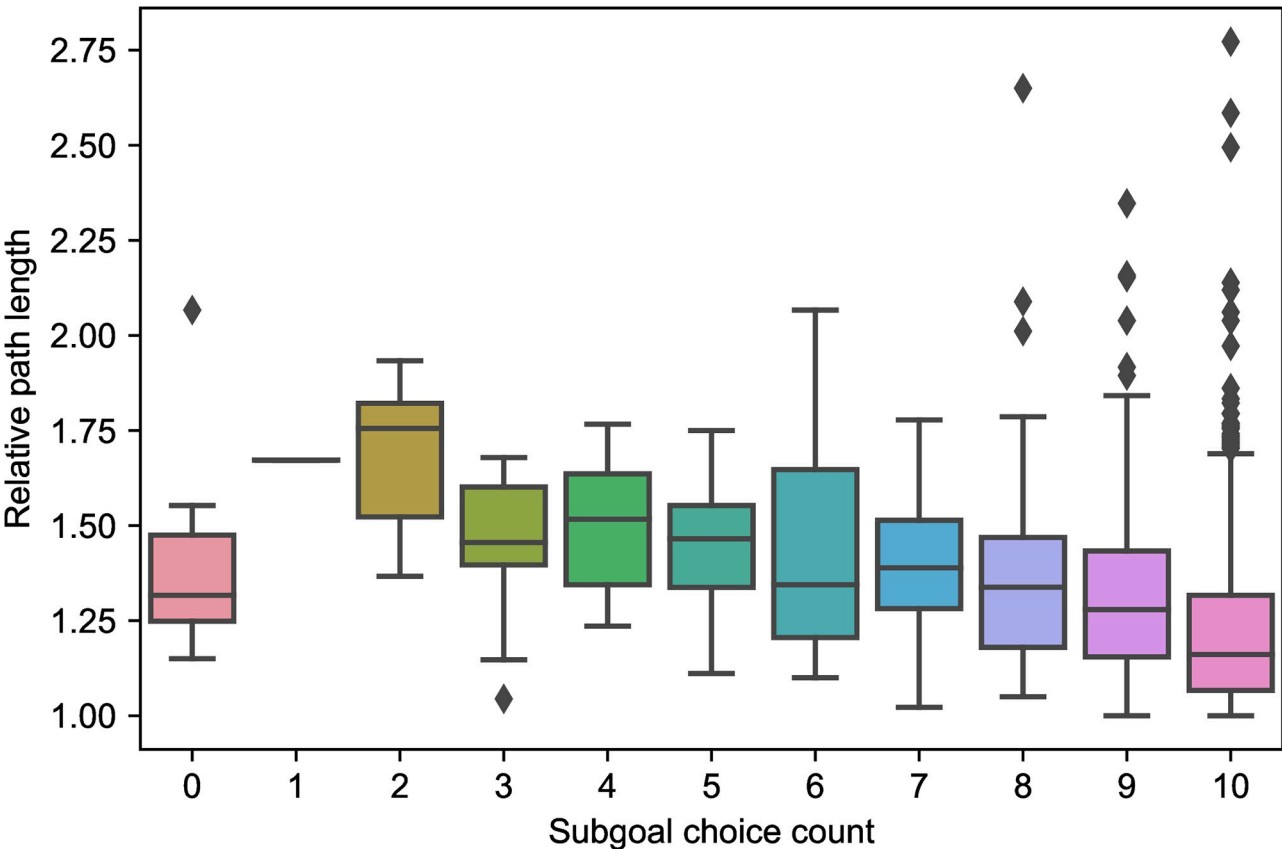

**Fig 6. Participants that choose a subgoal instead of the goal more often in Explicit Probe trials have shorter average path length, relative to the optimal path length ($r = -0.29$, $p < .001$).**

We also briefly examine the relationship between subgoal choice count on Explicit Probe trials and response times during navigation. We were unable to find evidence of a correlation between the subgoal choice count of a participant and their average log-transformed navigation trial duration ($r = 0.01$, $p = .858$). In order to understand how subgoal use influences response times, future studies should examine trial-level measures in appropriate experimental designs, an issue we remark on in the discussion.

To further link the probes to behavior, we examine instances where participants performed the same task (matched by start and goal state) in the navigation trials and the probe trials. This allows us to ask whether participants took paths that passed through the states they later identified as subgoals. To simplify the interpretation of the analysis, we focus on navigation trials where the participant's path was optimal and there were multiple optimal paths between the start and goal. Evaluated over these pairs of matched navigation and probe trials, we found that participants' choices on probe trials were consistent with their choices among optimal paths (Explicit Probe: 75.4%, Implicit Probe: 70.6%) more often than would be expected by random choice among optimal paths (Explicit Probe: 70.5%, $p < .001$; Implicit Probe: 65.2%, $p < .001$; Monte Carlo test). Probe trial choice is also a significant predictor of choice among optimal paths when analyzed using multinomial regression (Explicit Probe: $\chi^2(1) = 54.7$, $p < .001$, Implicit Probe: $\chi^2(1) = 62.5$, $p < .001$).

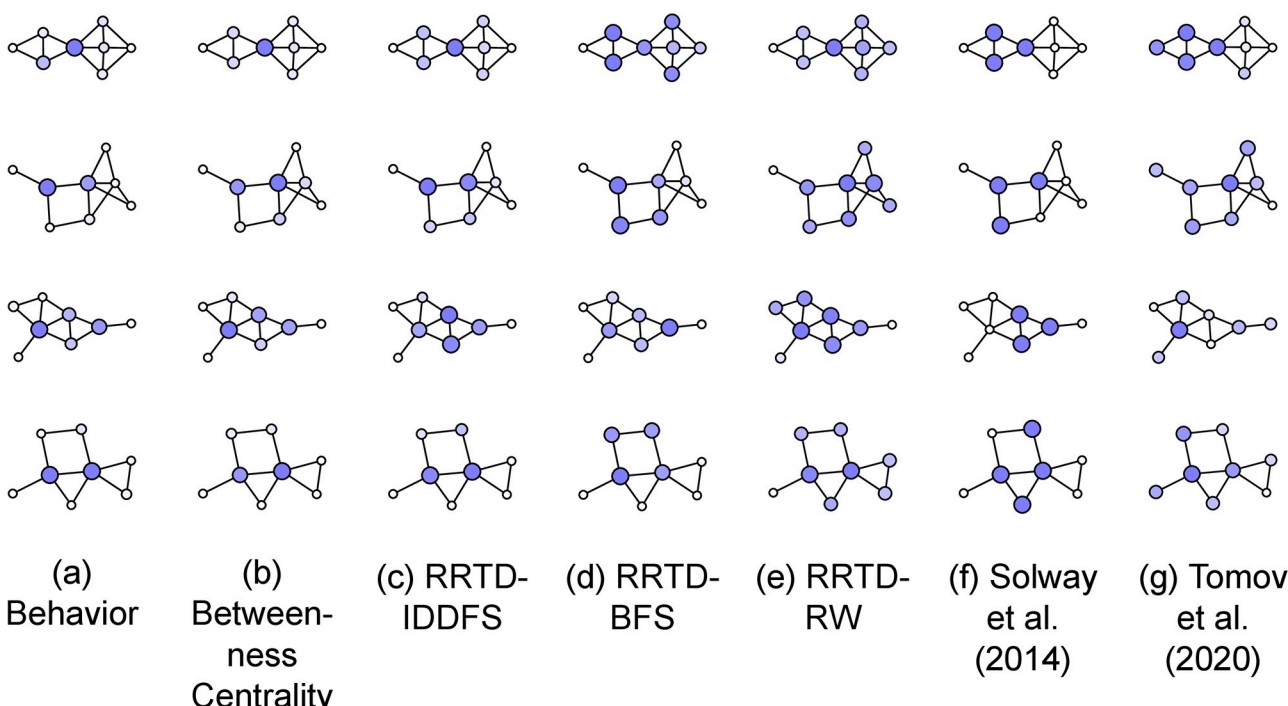

**Fig 7. Visualization of behavior and model predictions on four eight-node graphs selected from the 30 graphs used for the experiment.** (a) State color and size is proportional to subgoal choice, summed across participants and probe types. Each participant responded to a total of 21 subgoal probes and the number of participants per graph, from the top graph to the bottom graph, was 28, 26, 25, and 26. Visualized models are (b) RRTD-IDDFS, (c) RRTD-BFS, (d) RRTD-RW, (e) Betweenness Centrality, (f) Solway et al. (2014) [8], and (g) Tomov et al. (2020) [7]. For model predictions, state color and size is proportional to model prediction when using the state as a subgoal.

In sum, these results suggest the subgoal probes are well-correlated, though to a lesser degree for the Teleportation Question. They also suggest a strong connection between the probes and planning behavior.

**Comparing subgoal choice to theories.** Having established that participants learned the task, as well as the validity of their probe responses, we now turn to our central claim, namely that subgoal choice is driven by the computational costs of hierarchical planning. Letting participants' responses to the subgoal probes stand as reasonable proxies of subgoal choice, we relate the predictions of normative accounts and heuristics to participant probe choice across the three probes.

We start by qualitatively examining participant subgoal choice in Fig 7, extending the qualitative analysis given above with two additional graphs, more model predictions (Fig 7b–7g), and behavioral data averaged across tasks, probes, and participants (Fig 7a). Participant data for all graphs is in Fig A1 in S1 Appendix. We first note that Betweenness Centrality and RRTD-IDDFS are consistent in the graphs (Fig 7b and 7c), and are both relatively consistent with participant probe choice—as described above, states that are close to many other states are preferred. For brevity, we skip over RRTD-BFS and RRTD-RW in this description. The predictions of Solway et al. (2014) [8] are less consistent with participant probe choices (Fig 7f)—as previously described, the predictions are unintuitive because of the difficulty in mapping between partitions and subgoals, particularly when graph bottlenecks correspond to states instead of edges. The predictions of Tomov et al. (2020) [7] are also less consistent with participant probe choices (Fig 7g)—as previously described, the predictions do not correspond

with intuitive subgoals because the model relies on between-group edges being sparser than within-group edges.

We now quantitatively compare model predictions of participant subgoal choice.

For each model and probe type, we predict participant choices using hierarchical multinomial regression, where standardized model predictions are included as a factor with a fixed and per-participant random effect. Since the regression analyses have the same effect structure and the underlying theories being compared have no free parameters, we compare the relative ability of factors to predict probe choice through their log likelihood (LL) in Fig 8. We also report the results of likelihood-ratio tests to the null hypothesis of a uniformly random choice model in Table 2. As in the analysis above, we assume the task distribution is uniformly-distributed over all pairs of distinct states. Among normative theories, we found the RRTD-IDDFS model best explained behavior as judged by LL. For the Explicit and Implicit Probes, the next best models in sequence were RRTD-BFS, then both Solway et al. (2014) [8] and Tomov et al. (2020) [7] with similar performance, and finally RRTD-RW. For the Teleportation Question, the best models after RRTD-IDDFS were Solway et al. (2014) [8], Tomov et al. (2020) [7], RRTD-BFS, and finally RRTD-RW. This suggests that, among the normative theories, those based on search costs are most explanatory of subgoal choices.

We additionally compare model predictions to participant state occupancy during navigation trials in order to assess whether people are relying on simple, memory-based strategies to respond to the probes, as described above. We find that participant behavior is better explained by all normative theories for the Explicit and Implicit Probes (with the exception of RRTD-RW), but only RRTD-IDDFS and Solway et al. (2014) [8] for the Teleportation Question.

Among the heuristic theories, we found Betweenness Centrality best explained behavior as judged by LL. For all probes, Degree Centrality was next best, followed by QCut. As above, we compared model predictions to participant state occupancy and found that participant behavior is better explained by Betweenness Centrality for the Explicit and Implicit Probes, and both Betweenness and Degree Centrality for the Teleportation Question.

These results are consistent with the empirical connection between RRTD-IDDFS and Betweenness Centrality found in the large-scale simulation above. Betweenness Centrality further improves on the behavioral fit of RRTD-IDDFS, suggesting that our participants may be using a metric like Betweenness Centrality to approximate resource-rational task decomposition. In contrast, we found that state occupancy was a worse fit to participant behavior than either RRTD-IDDFS or Betweenness Centrality, suggesting that the introduction of filler trials was sufficient to rule out a trivial strategy based on state occupancy. We return to these points in the discussion.

Since the experiment was designed so that only local connections were visible during navigation trials but conducted via an online platform, in the closing survey we asked participants if they used a reference to the task structure besides the interface ("Did you draw or take a picture of the map? If you did, how often did you look at it?"). Participant responses were as follows: 603 participants selected "Did not draw/take picture," 65 selected "Rarely looked," 90 selected "Sometimes looked," and 48 selected "Often looked." In order to ensure the above results were not impacted, we ran the same analysis in the subset of participants ($N = 603$) that selected "Did not draw/take picture" and found qualitatively similar results (Fig A2 and Table A1 in S1 Appendix).

In another analysis in S1 Appendix, we tested whether icon identity influenced these results by incorporating the icon used for state presentation into the null choice model. We found minimal influence—like the above results, the addition of subgoal predictions was statistically significant for each model and comparisons based on log likelihood were qualitatively similar.

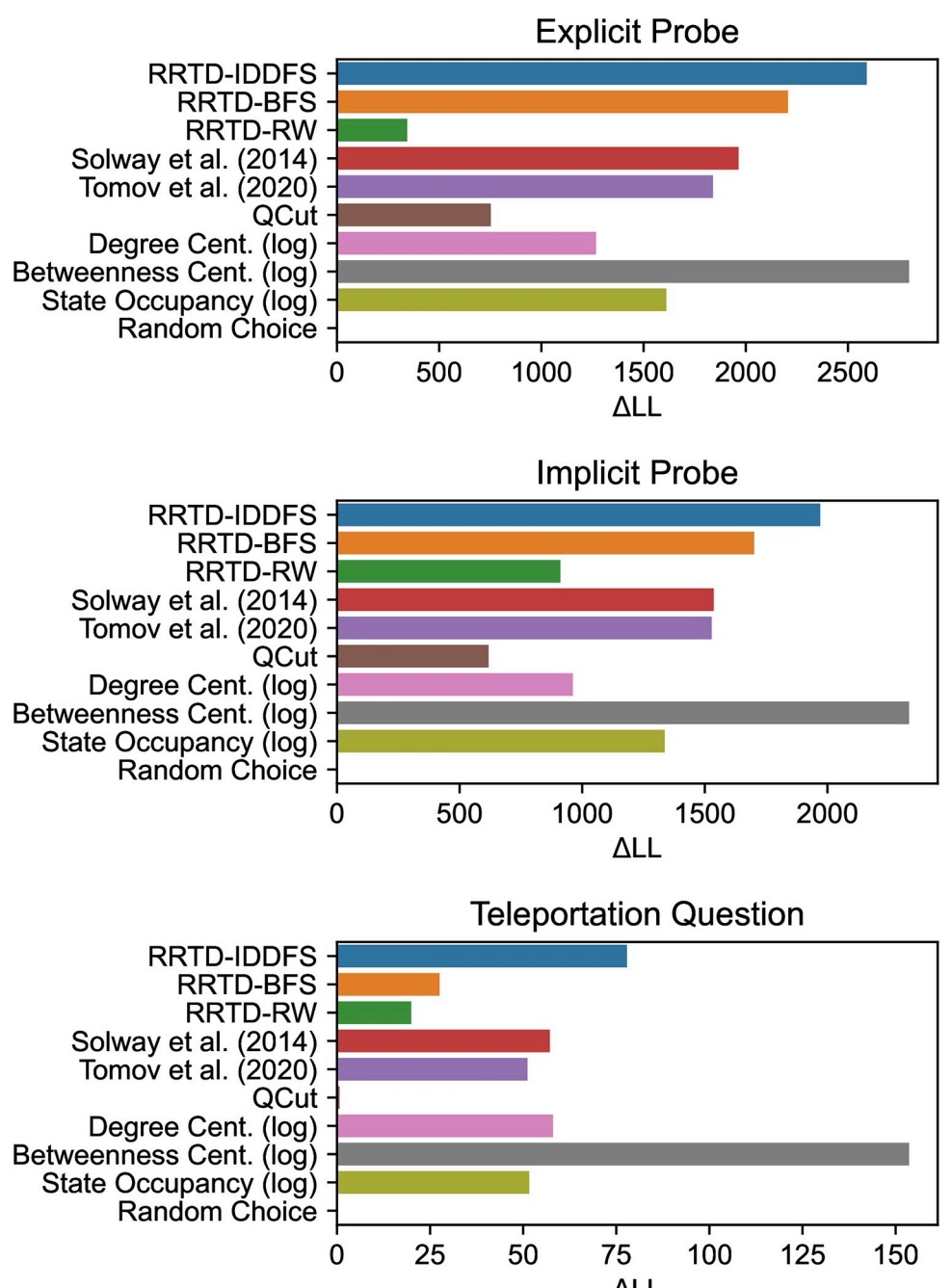

**Fig 8. Comparison of statistical analysis using mixed-effects multinomial regression to predict subgoal choice behavior for each subgoal probe.** Log likelihood (LL) is relative to the minimum model LL for each probe. Larger values indicate better predictivity. References: Solway et al. (2014) [8], Tomov et al. (2020) [7].

In a final analysis, we predict participant navigation in the instances where their path was one of several optimal paths. We analyze participant choice as a simple two-stage process: a subgoal is sampled with log probability proportional to model predictions (weighted by a parameter $\beta_1$), then an optimal path is sampled with log probability proportional to a free parameter $\beta_2$ if it contains the subgoal and 0 otherwise. For each theory of subgoal choice, we

**Table 2. Estimated coefficients with standard errors from hierarchical multinomial regression predicting subgoal choice.** Likelihood-ratio test statistics compare regression models to the null hypothesis of sampling subgoals uniformly at random.

| Normative Algorithm | Explicit Probe | Implicit Probe | Teleportation Question |
|---|---|---|---|
| RRTD-IDDFS | $\beta = 1.78$ | $\beta = 1.63$ | $\beta = 0.73$ |
| | $SE = 0.04$ | $SE = 0.04$ | $SE = 0.06$ |
| | $\chi^2(2) = 5183.1$ | $\chi^2(2) = 3941.2$ | $\chi^2(1) = 155.7$ |
| | $p < .001$ | $p < .001$ | $p < .001$ |
| RRTD-BFS | $\beta = 4.98$ | $\beta = 4.94$ | $\beta = 1.45$ |
| | $SE = 0.10$ | $SE = 0.11$ | $SE = 0.20$ |
| | $\chi^2(2) = 4412.8$ | $\chi^2(2) = 3402.8$ | $\chi^2(1) = 55.1$ |
| | $p < .001$ | $p < .001$ | $p < .001$ |
| RRTD-RW | $\beta = 0.37$ | $\beta = 0.92$ | $\beta = 0.29$ |
| | $SE = 0.02$ | $SE = 0.03$ | $SE = 0.05$ |
| | $\chi^2(2) = 686.7$ | $\chi^2(2) = 1822.1$ | $\chi^2(1) = 39.9$ |
| | $p < .001$ | $p < .001$ | $p < .001$ |
| Solway et al. (2014) [8] | $\beta = 0.75$ | $\beta = 0.69$ | $\beta = 0.37$ |
| | $SE = 0.02$ | $SE = 0.02$ | $SE = 0.03$ |
| | $\chi^2(2) = 3929.7$ | $\chi^2(2) = 3072.7$ | $\chi^2(1) = 114.3$ |
| | $p < .001$ | $p < .001$ | $p < .001$ |
| Tomov et al. (2020) [7] | $\beta = 1.09$ | $\beta = 0.97$ | $\beta = 0.41$ |
| | $SE = 0.03$ | $SE = 0.02$ | $SE = 0.04$ |
| | $\chi^2(2) = 3678.7$ | $\chi^2(2) = 3056.4$ | $\chi^2(1) = 102.4$ |
| | $p < .001$ | $p < .001$ | $p < .001$ |
| **Heuristic** | **Explicit Probe** | **Implicit Probe** | **Teleportation Question** |
| QCut | $\beta = -0.14$ | $\beta = -0.19$ | $\beta = 0.04$ |
| | $SE = 0.01$ | $SE = 0.01$ | $SE = 0.04$ |
| | $\chi^2(2) = 1504.3$ | $\chi^2(2) = 1236.3$ | $\chi^2(1) = 1.4$ |
| | $p < .001$ | $p < .001$ | $p = .238$ |
| Degree Cent. (log) | $\beta = 0.73$ | $\beta = 0.64$ | $\beta = 0.45$ |
| | $SE = 0.02$ | $SE = 0.02$ | $SE = 0.04$ |
| | $\chi^2(2) = 2534.3$ | $\chi^2(2) = 1923.3$ | $\chi^2(1) = 116.0$ |
| | $p < .001$ | $p < .001$ | $p < .001$ |
| Betweenness Cent. (log) | $\beta = 0.86$ | $\beta = 0.82$ | $\beta = 0.58$ |
| | $SE = 0.02$ | $SE = 0.02$ | $SE = 0.03$ |
| | $\chi^2(2) = 5598.2$ | $\chi^2(2) = 4666.1$ | $\chi^2(1) = 307.3$ |
| | $p < .001$ | $p < .001$ | $p < .001$ |
| State Occupancy (log) | $\beta = 1.00$ | $\beta = 0.89$ | $\beta = 0.39$ |
| | $SE = 0.03$ | $SE = 0.03$ | $SE = 0.04$ |
| | $\chi^2(2) = 3222.8$ | $\chi^2(2) = 2673.3$ | $\chi^2(1) = 103.3$ |
| | $p < .001$ | $p < .001$ | $p < .001$ |

optimized this two-stage choice model to maximize the likelihood assigned to observed choices. Because there were a relatively small number of trials per participant, we did not fit random effects for participants. The model results are shown in Fig 9. These results are again consistent with those previously observed—among normative theories RRTD-IDDFS is best, among heuristic theories Betweenness Centrality is best, and Betweenness Centrality is overall the most explanatory. In a supplementary analysis, we also found evidence that participant responses during optimal navigation were slower at both self-reported subgoals and those predicted by models, as detailed in S1 Appendix.

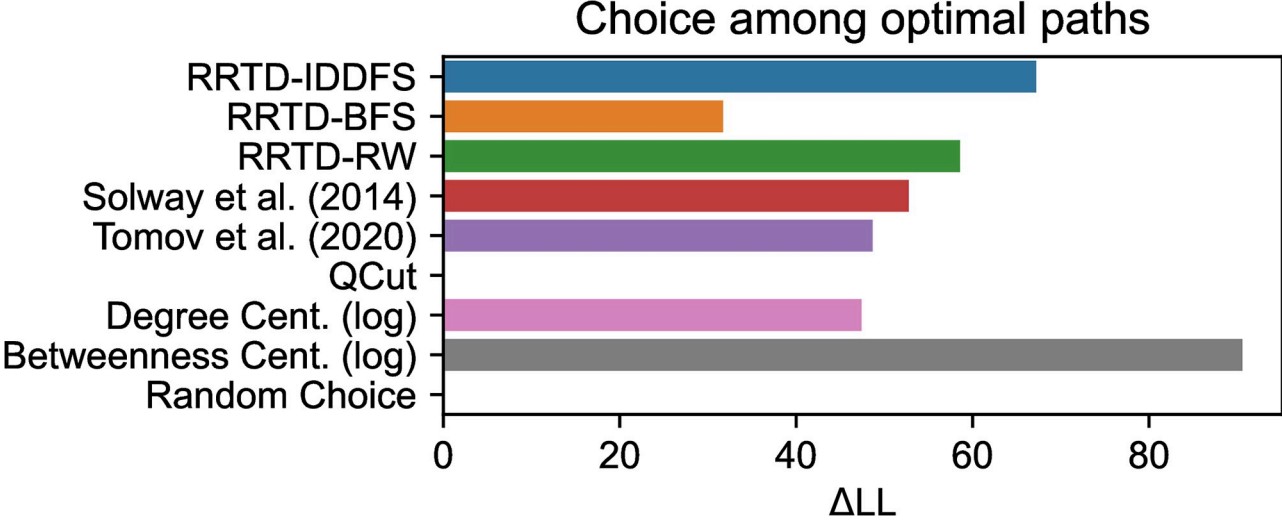

**Fig 9. Comparison of two-stage choice models to predict participant choice behavior among optimal paths.** Log likelihood (LL) is relative to the minimum model LL. Larger values indicate better predictivity. References: Solway et al. (2014) [8], Tomov et al. (2020) [7].

## Discussion

In this work, we have proposed a resource-rational framework for task decomposition where tasks are broken down into subtasks based on planning costs. Our first contribution is a novel formal account of this idea based on a resource-rational analysis [38]. Specifically, our proposal involves three levels of nested optimization: *Task decomposition* identifies a set of subgoals for a given task, *subtask-level planning* chooses sequences of subgoals to reach a goal, and *action-level planning* chooses sequences of concrete actions to reach a subgoal. Optimal task decomposition thus depends on both the structure of the environment and the computational resource usage specific to the planning algorithm. We quantitatively compared the predictions of our framework to four heuristic and normative theories proposed in the literature across 11,117 graph-structured tasks. These analyses show that our framework provides different predictions from other normative accounts and aligns with heuristics. We argue that this provides a rationalization of these heuristics for task decomposition in terms of resource-rational planning that accounts for computational costs.

To test our framework, we ran a pre-registered, large-scale study using 30 graph-structured environments and 806 participants. This study includes a more diverse set of tasks than that of any previously reported study in the literature, allowing us to draw more general conclusions about how people form task decompositions. The results of this study reveal that, among normative models, people's responses most closely align with the predictions of our model. This provides support for our theory that people are engaged in a process of resource-rational task decomposition. Among heuristics for task decomposition, one heuristic is a better fit to behavior than our framework. Because the heuristic makes similar predictions as our framework, this might indicate that people use the heuristic as a tractable approximation to our framework.

Our account, while normative, is not particularly interpretable. Identifying the qualitative patterns that guide human subgoal choice and relating them to the patterns resulting from our framework's sensitivity to search costs will be necessary for an interpretable account of human subgoal choice. Critically necessary are experimental paradigms that provide rich, but minimally confounded behavior—our experiments extend those in the literature, but our results

depend heavily on the self-reported subgoals of research participants. While we have already demonstrated relationships between self-reported subgoals and behavior, making more extensive comparisons to behavior is important for future research. For example, although we found a systematic relationship between participants' responses to the subgoal probes and their previous navigation decisions, these two measurements were taken minutes apart. We chose to separate these two measurements so as to avoid possible measurement effects in which explicitly asking about subgoals may lead people to navigate through the state they identified. However, this likely weakens the observed relationship between the probe responses and the navigation decisions. An additional difference during navigation trials is that participants are still learning the task structure and are shown connections from the current state. By contrast, during the probe trials, participants are unable to see any connections and must rely solely on what they have learned. These subtle differences may reduce the relationship between these two trial types. They are also different from the formal framework which assumes perfect knowledge about task structure—because this could influence task decomposition, we note a relevant extension below. The effect of this difference could be studied by introducing a separate experimental phase where participants are trained on the task structure directly, outside the context of navigation—the first experiment in Solway et al. (2014) [8] contains a similar phase, but had limited navigation trials. Developing experimental techniques to measure subgoal choices in the process of navigation without biasing the planning process is an important direction for future work.

We highlight three limitations of the navigation trials studied in this experiment. The first is that participants are encouraged to plan hierarchically ("It might be helpful to set subgoals" in Fig 5a). While this seems likely to have minimal impact on *which* subgoals participants choose, the main focus of this manuscript, it may impact *whether* participants plan hierarchically in the first place. Future studies intending to assess how people choose to plan hierarchically should consider avoiding prompts like this.

Second, though we report analyses of participant response times, these analyses were not pre-registered and our experiment was not designed to assess response times. These findings should be reevaluated in experimental designs appropriate to assess response times. For example, we found that participants were slower to respond at their subgoals, suggesting they were planning at those states. However, our normative framework makes no prediction about *when* planning occurs. The framework only defines a resource-rational value for subgoals that can be used to simplify planning whether it happens before action or after reaching a subgoal. Future experiments could investigate this further through manipulations to influence when planning is employed, like a timed phase for up-front planning or an incentive for fast plan execution.

A third limitation is that the navigation trials provide limited insight into the algorithms people are using to plan. While participants only see local connections during navigation, analogous to the local visibility search algorithms have, there are a number of reasons why their navigation behavior would be difficult to relate to the choice of search algorithm. The main issue is that planning steps are generally covert, and do not correspond in any simple way with steps of overt behavior given by the plan that is ultimately produced. Such covert planning is better suited, in future work, to being studied through process-tracing experiments [30, 39], think-aloud protocols, or by investigating neural signatures related to planning and learning [40]. It is possible that some aspects of planning are externalized in the current experiment, via exploration on navigation trials, but this is at best incomplete. For instance, participant behavior improves over the course of navigation trials, suggesting they are performing mental search instead of search via navigation. Also, the experiment restricts single-step movement to neighboring states, whereas by contrast, algorithms like BFS might plan over states in an order

where subsequent states are not neighbors. Identifying the search algorithm participants use is critical for future studies since our framework predicts that task decomposition is driven by the search algorithm used.

As mentioned in the text, our framework and experiment explicitly focus on the constrained setting of bruteforce search. However, other theoretical studies have extended this framework to incorporate heuristic search [12, 32] and abstract subgoals [32]. Future research should continue to explore extensions of this framework to more robustly test the predictions of resource-rational task decomposition. For example, our framework could be used to make predictions about subgoal choice in spatial navigation tasks by incorporating spatial distance heuristics and using heuristic search algorithms like A* search or Iterative-Deepening A* search [25]. Another direction could explore other resource costs like memory use, motivated by the relatively low memory use of IDDFS discussed above. At present, our framework assumes planning for tasks occurs independently, avoiding the reuse of previous solutions to subtasks. Despite this absence, our framework still predicts a normative benefit for problem decomposition. However, research has found that people learn hierarchically, exhibiting neural signatures consistent with those predicted by hierarchical reinforcement learning theory [41]. Further research could relax the independence between task solutions by explicitly reusing solutions (as in [8]) or turning to formulations based on reinforcement learning [42], particularly those designed for a distribution of goal-directed tasks with shared structure as studied in this manuscript [43]. A particularly interesting direction could incorporate model learning (as in the Dyna architecture [44]) across tasks in order to explain the influence of task learning on task decomposition. Further extension of this framework could build on resource-rational models developed in other domains, like Markov Decision Processes [19] and feedback control [45], and draw inspiration from approaches used to learn action hierarchies in high-dimensional tasks [42, 43].

Our framework is intended to be an idealized treatment of the problem of task decomposition, but an essential next step for this line of research is to understand how people tractably approximate the expensive computations needed to determine search costs, which we discuss above. Our results already make some progress in this direction. Specifically, we found that participant responses were best explained by the Betweenness Centrality model. This model's predictions are highly correlated with the normative RRTD-IDDFS model but require far less computation to produce. This suggests that people may be using Betweenness Centrality as a heuristic to approximate the task decomposition that minimizes planning cost. However, Betweenness Centrality is also expensive to compute since it requires finding optimal paths between all pairs of states–something our participants are not likely doing. Since we were able to rule out one trivial estimation strategy (state occupancy) through the inclusion of filler trials, the issue of tractable estimation is an open question for future research to explore by proposing other estimation strategies and experimental manipulations to dissociate their predictions. Identifying even more efficient approximations to resource-rational task decomposition will be essential for a process-level account of human behavior, as well as for advancing a theory of subgoal discovery for problems with larger state spaces.

The human capacity for hierarchically structured thought has proven difficult to formally characterize, despite its intuitive nature and long history of study [13, 28]. In this study we propose a resource-rational framework that motivates and explains the use of hierarchical structure in decision-making: People are modeled as having subgoals that reduce the computational overhead of action-level planning. Our framework departs from and complements other normative proposals in the literature. Most published accounts pose task decomposition as an *inference problem*: People are modeled as inferring a generative model of the environment [6, 7] or as compressing optimal behavior [8, 20]. We quantitatively relate our

framework to existing proposals in a simulation study; In addition, we conduct a large-scale behavioral experiment and find that our framework is effective at predicting human subgoal choice. The work presented here is consistent with other recent efforts within cognitive science to understand how people engage in computationally efficient decision-making [9–11, 38]. It is also complementary to recent work in artificial intelligence that explores the interaction between planning and task representations [19, 42]. Our hope is that future work on human planning and problem-solving will continue to investigate the relationships between computation, representation, and resource-rational decision-making.

## Methods

### Ethics statement

The following experimental procedures were approved by the institutional review board of Princeton University. In the experiment, participants were shown an electronic consent form providing a written description of the experiment and gave informed consent by clicking a button in lieu of a signature.

### Experiment design

To probe for participant subgoal choice, we employed an experiment inspired by those previously published in Solway et al. [8]. In our experiment, participants first navigated on a web-like representation of a graph to learn the graph's structure ("navigation trials"; Fig 5), then answered a series of task-specific and task-independent questions about subgoal choices ("probe trials"). We then quantitatively analyzed their responses to these questions about subgoal choice. In the Design section, we motivate the choice of various experimental details. Then, in the Procedure section we detail the experimental procedure. The experiment pre-registration is available at https://osf.io/hegf2. Our pre-registered analysis was a comparison of how well RRTD-IDDFS and Solway et al. (2014) [8] could predict participant probe choice using hierarchical multinomial regression, a subset of the comparisons in Fig 8 and Table 2.

**Design.**   The navigation trials were intended to provide participants with an opportunity to learn the structure of the graph. Drawing from results in pilot experiments, we ensured participants could only see the graph edges connected to their current state (Fig 5a). Periodically, the graph with all edges was shown to participants (Fig 5b). Importantly, this was done without signaling any information about future tasks. From pilot experiments, these visual choices (minimizing displayed edges during tasks, but showing all edges periodically between tasks) ensured that participants quickly learned the graph structure instead of relying on the visual representation of the graph.

From pilot experiments, we observed that states with high visit rates coincided with the predictions of RRTD-IDDFS and Betweenness Centrality. To address this confound, the experiment had two types of navigation trials: long and filler. Long trials were optimally solved with more than one action (i.e. the start and goal state were not directly connected) and were intended to give participants exposure to the graph structure. Filler trials were optimally solved with one action (i.e. the start and goal state were directly connected) and were adaptively chosen to ensure a balanced visit rate that avoided overlapping predictions with our model. This adaptive procedure selected from all possible filler tasks by 1) excluding tasks where the start or goal state was most-visited, and then by 2) sampling uniformly from the remaining tasks with the greatest sum of visits to the start and goal states. When all tasks had a most-visited state as a start or goal, the first step was skipped; this circumstance was uncommon and dependent on both participant behavior and the structure of the graph. In effect, this procedure increased the number of most-visited states by increasing visits to states that were nearly (but

not) most-visited. In simulations and pilot experiments, this procedure was sufficient to dissociate the visit rate and our model predictions.

Our probes for subgoal choice included two inspired by prior studies [8]—the implicit probe and teleportation question—and included a novel variant that explicitly asked about subgoal use—the explicit probe. We included these three probes to ensure a reliable and comprehensive measure of subgoal choice. The implicit probe was shown to participants before the explicit probe to avoid biasing participant responses.

Typical visual layouts of graphs often have a relationship to pairwise state distances, which means that a variety of visual heuristics are effective strategies when problem-solving. To avoid these confounding heuristics, we visually represented the graph states in a pseudo-random circular layout (Fig 5). The same circular layout was shown for all trial types, including the periodic display of all graph edges.

**Procedure.**   Each participant was assigned a single graph for the entire experiment, with a fixed circular layout and fixed mapping from nodes to icons. We first introduced them to the experimental interface with an interactive tutorial, showing the visual cues used to mark the current node and goal node and how to navigate along graph edges using the numeric keys of the keyboard. We then introduced the concept of a "subgoal" through the example of a cross-country road trip with a "subgoal" located at the midpoint of the road trip. Participants were asked to describe what they thought "subgoal" meant.

Navigation trials followed this introductory material. Participants completed a few short practice trials, then completed 60 trials alternating between long and filler trial types: 30 long trials were drawn uniformly from those optimally solved with more than one action, and 30 filler trials were adaptively selected from those optimally solved with one action (described in detail above). In navigation trials, participants started from a node and had to navigate to a goal node. They were also prompted to consider the use of subgoals with the message "It might be helpful to set subgoals." At any point during a navigation trial, participants could only see the edges connected to their current node (Fig 5a). Every four trials (thus, 15 times total) participants were shown all the edges of the graph (Fig 5b). Since a photograph of the graph shown this way could simplify navigation trials, the icons at each node were only shown when the participant hovered over them.

Following navigation trials, we probed for subgoal choice. For all probes, the graph was shown in the same circular layout, but the edges were hidden. Between every two trials, the graph was shown with all edges as mentioned above. In the context of 10 different tasks, we queried for subgoal choice using the implicit probe: "Plan how to get from A to B. Choose a location you would visit along the way." For this probe, we excluded both the start and goal nodes from the available options. Then, in the context of the same 10 tasks after shuffling, we queried for subgoal choice using the explicit probe: "When navigating from A to B, what location would you set as a subgoal? (If none, click on the goal)." For this probe, we only excluded the start node from the available options. Before each of the task-specific probes, participants also completed a few practice trials. Finally, we asked a single instance of the teleportation question: "If you did the task again, which location would you choose to use for instant teleportation?" For this probe, all nodes were available options.

Finally, participants responded to a multiple-choice survey question: "Did you draw or take a picture of the map? If you did, how often did you look at it?"

**Stimuli.**   The graphs we studied were sampled from among the 11,117 simple, connected, undirected, eight-node graphs with sufficient probe trials for the study design. For a given graph, probe trials were sampled uniformly from tasks that require at least 3 actions to optimally solve, a threshold selected based on model predictions that these tasks often require the use of hierarchy. After ensuring each graph had 10 distinct tasks that require 3+ actions to

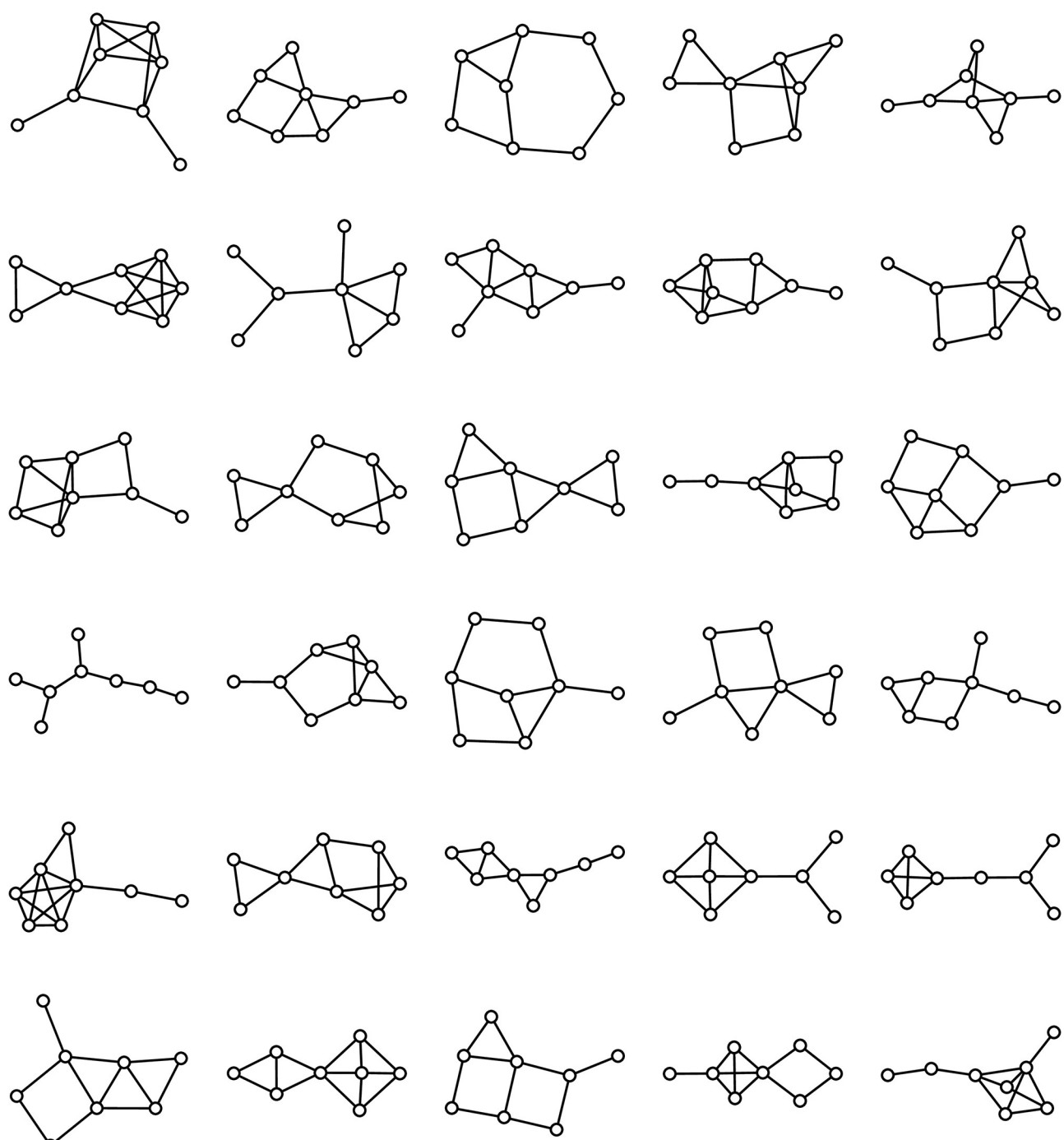

**Fig 10. The 30 undirected, eight-node graphs that were used in the experiment.**

optimally solve, this limited the number of possible graphs to 1,676. The 30 graphs we studied were sampled uniformly at random from these 1,676 graphs (Fig 10). The order of graph nodes in the circular layout, the icon assigned to each node, and the sequence of navigation and probe trials were all sampled pseudo-randomly. We counter-balanced the assignment of participants to graph, circular layout, and trial orderings.

All icons designed by OpenMoji, the open-source emoji and icon project. License: CC BY-SA 4.0.

## Analyses

**Model predictions.**   We define a model's predictions over subgoals as proportional to a utility or log probability. We do so because model predictions are primarily used to predict probe choice using multinomial regression. This leads to a natural interpretation of the coefficients from multinomial regression. For a utility, multinomial regression is equivalent to a softmax choice model, so the coefficient can be equivalently interpreted as an inverse temperature. For a log probability, the coefficient $w$ can predict a range of strategies, including random choice for $w = 0$, probability matching for $w = 1$, and maximizing based on probability as $w \to \infty$.

We define the task distribution, when applicable to a model, as a uniform distribution over all tasks with distinct start and goal states. The reported analyses have qualitatively similar results when the task distribution matches the experiment's long trials, which only includes tasks $(s_0, g)$ where the start $s_0$ and goal $g$ are not neighbors, so $(s_0, g) \notin T$.

**Hierarchical multinomial regression of choice.**   We model participant choice among subgoals using hierarchical multinomial regression with the `mlogit` package in the R programming language. Regression models are fit with 100 draws from the default Halton sequence (parameters `halton = NA, R = 100`).

For each model, we predict participant choices for each type of probe trial using multinomial regression with model predictions as regressors. Regressors were standardized to be mean-centered with a standard deviation of 1. Since each participant has multiple task-specific probe choices for the explicit and implicit probes, we include random effects for regressors when modeling those probe types. While not explicitly noted above, since the teleportation question was only asked once per participant, prediction of subgoal choice for it was fit without random effects (i.e. non-hierarchical multinomial regression).

For task-specific probes, the set of possible choices available to the model are configured to match those available to participants, as described in the Procedure section. Additionally, the explicit probe instructs participants to select the goal if they did not use subgoals. For RRTD-based models, we model this with the predicted value for the use of no subgoals. This is equivalent to the sum of the reward and negated planning cost of navigating directly to the goal.

**Two-stage model of choice among optimal paths.**   Free parameters $\beta_1$ and $\beta_2$ were constrained to be greater than or equal to zero. Optimization started from initial parameters $\beta_1 = 1, \beta_2 = 1$. The random choice model selects subgoals uniformly at random which corresponds to a special case of the two-stage choice model with fixed parameter $\beta_1 = 0$.

## Resource-rational task decomposition

The model predictions for a state $s$ are the value of a task decomposition (Eq 3) where the state is a single subgoal, $V(\mathcal{Z}) = V(\{s\})$.

**Random walk.**   The search algorithm returns a plan $\pi = \langle s_0, s_1, \ldots, z \rangle$ and run-time $t = |\pi|$ with probability $P_{\text{RW}}(\pi, t \mid s, z) = \prod_{i>0} \frac{1}{\mathcal{N}(s_i)}$. Since we defined the reward for a plan as $R(\pi) = -|\pi|$, the expected reward over all plans is

$$R_{\text{RW}}(s, z) \quad = \sum_{\pi, t} P_{\text{RW}}(\pi, t \mid s, z)[R(\pi) - t]$$

$$= -2 \sum_{\pi, t} P_{\text{RW}}(\pi, t \mid s, z)|\pi|.$$

Since a constant multiplier does not affect model predictions, we drop the constant and let $R_{\mathrm{RW}}(s, z) = -\Sigma_{\pi,t} P_{\mathrm{RW}}(\pi, t \mid s, z)|\pi|$.

The negative expected reward $-R_{\mathrm{RW}}(s, z)$ is the expected number of steps until the first visit to $z$ when starting at $s$, also called the hitting time $H(s, z)$. $H(s, z)$ can be efficiently computed by a recursive equation

$$H(s, z) = 1 + \sum_{s' \in \mathcal{N}(s)} \frac{1}{|\mathcal{N}(s)|} H(s', z)$$

when $s \neq z$, and with $H(s, s) = 0$ otherwise. We thus define $R_{\mathrm{RW}}(s, z) = -H(s, z)$.

Since the use of subgoals will either maintain or increase the number of steps required to reach a goal for a random walk, we make note of implementation differences to accommodate this for RRTD-RW. Formally stated, $H(s, s') + H(s', s'') \geq H(s, s'')$, with $H(s, s') + H(s', s'') = H(s, s'')$ only when all state sequences from $s$ to $s''$ must contain $s'$. So, states $s'$ with $H(s, s') + H(s', s'') > H(s, s'')$ will only increase the expected number of steps and would not be in the policy over subgoals defined by Eq 2. To avoid this issue, we require the subgoal policy for RRTD-RW to contain at least one subgoal. By the same argument, a second subgoal can not decrease the expected number of steps, so we can simplify Eq 2 for RRTD-RW to

$$V_{\mathcal{Z}}^g(s) = \max_{z \in \mathcal{Z}}\{-[H(s, z) + H(z, g)]\} \tag{4}$$

or when the subgoals are a singleton set $\mathcal{Z} = \{z\}$ simply $V_{\mathcal{Z}}^g(s) = -[H(s, z) + H(z, g)]$.

**Depth-first search.** The algorithm is recursively defined to take a current state $s$ and plan-so-far $\pi$. At each call of the algorithm, it iterates over neighbors of the current state $s' \in \mathcal{N}(s)$—if the state $s'$ is not in the current plan $\pi$ then there is a recursive call to the algorithm with state $s'$ and an updated plan $\pi'$ that ends with $s'$. When there are no unvisited neighbors $s'$ to consider, the algorithm backtracks to a previous state and plans until it finds one with unvisited neighbors. When the algorithm reaches the subgoal $z$, it terminates, returning the plan and a run-time based on the number of calls to the algorithm. To avoid bias due to neighbor order, in each call of the algorithm neighbors are randomly shuffled.

**Breadth-first search.** The algorithm has a queue of states to visit and tracks all states that have been visited. At each iteration, it visits the next state $s$ from the queue and adds all not-yet-visited neighbors $s' \in \mathcal{N}(s)$ to the queue. When it visits the subgoal $z$, it returns the path to $z$ and a run-time based on the number of iterations that were required. As in DFS, we shuffle the neighbors of the current state at each iteration to avoid bias due to neighbor order.

**Iterative deepening depth-first search.** A depth-limited DFS augments a standard DFS by terminating when the current "depth" (i.e. the length of the current plan) exceeds a limit, in addition to terminating when the goal is reached. IDDFS iterates by running depth-limited DFS with incrementally larger depths, starting from a depth limit of 1. The algorithm returns when a depth-limited DFS finds a plan to the goal, counting the run-time as the number of recursive DFS calls across all uses of depth-limited DFS. As in other algorithms, neighbors are shuffled to avoid bias due to order.

## Alternative models

**Degree centrality, betweenness centrality.** Both centrality measures were computed in the Python programming language using the `networkx` library with all parameters left at their defaults except for `endpoints = True` for Betweenness Centrality. As computed by `networkx`, both centrality measures are a fraction—for a given state $s$, Degree Centrality is proportional to the fraction of states that $s$ is connected to and Betweenness Centrality is the

fraction of optimal paths that $s$ is part of. Thus, for both measures, we define the model predictions as the logarithm of these fractions for the reasons noted above.

**QCut.** This section requires more extensive use of graph theory, so we first explicitly connect the task formalism used in the remainder of the text to graphs before describing QCut. An undirected graph consists of nodes $\mathcal{V}$ and edges $\{i, j\} \in \mathcal{E}$ between nodes $i$ and $j$, and we let $n = |\mathcal{V}|$ and $m = |\mathcal{E}|$. The adjacency matrix of a graph $A_{ij} = 1$ if $\{i, j\} \in \mathcal{E}$ and 0 otherwise. The degree of a node $i$ is $d_i = \sum_j A_{ij} = \sum_i A_{ij}$ and the degree matrix $D = diag(d)$. To connect the notation used in the rest of the paper, we contextually assume the following relationship between undirected graphs and task environments: For environments with reversible actions (formally, $(s, s') \in T \Leftrightarrow (s', s) \in T$), we let states correspond directly to graph nodes and transitions in the environment $(s, s') \in T$ correspond to graph edges $\{s, s'\} \in \mathcal{E}$.

QCut divides the states of a graph into two groups based on the Normalized Cut criterion, which admits an approximate solution based on a spectral decomposition of the graph [16, 33, 46]. The approximate solution to the Normalized Cut criterion is based on the symmetric graph Laplacian $\mathcal{L}_{\text{sym}} = D^{-\frac{1}{2}}(D - A)D^{-\frac{1}{2}} = I - D^{-\frac{1}{2}}AD^{-\frac{1}{2}}$.

Following prior work [16, 46], we divide graph nodes into two groups based on the eigenvector $v$ with the second-smallest eigenvalue of $\mathcal{L}_{\text{sym}}$. This eigenvector is also the best, non-trivial one-dimensional embedding of the graph states that minimizes the distance between connected states (See Eq. 10 in [46]). Our implementation partitions states based on whether they are above or below a threshold in this one-dimensional embedding $v$—a typical threshold is zero or the median. Since states $s$ with corresponding eigenvector entry $v_s$ closest to the eigenvector mean are considered most central, we define the model prediction for a state $s$ as $-v_s^2$.

**Solway et al. (2014).** Solway et al. (2014) [8] propose that people choose hierarchies that most efficiently encode problem-solving behavior. The efficiency of an encoding is quantified through the information-theoretic concept of minimum description length; when applied to encode problem-solving behavior through hierarchical structure, this involves choosing a task decomposition so that solutions have short description length and are composed of subtasks whose solutions can be reused in many tasks. This account takes optimal paths as the behavior to encode and selects hierarchies based on graph partitions.

We now note our implementation details that depart from those of Solway et al. (2014) [8]. To predict behavioral choices, we use the optimal behavioral hierarchy to specify a binary regressor that takes a value of 1 for *boundary states* (states with at least one neighbor in a different region) and 0 otherwise—we discuss this choice in the main text. Since multiple state sequences can be optimal, random noise is added to graph edges for the purpose of tie-breaking—in our implementation, the description length of behavior is averaged across 10 samplings of these edge weights in order to reduce the effects of noise. In the original publication, optimization over partitions based on model evidence was performed using a genetic algorithm—in our implementation, we enumerate all graph partitions and select those with the highest model evidence. For a given graph, the original article considers all possible partitions—for our analyses over eight-node graphs, we found it necessary to exclude trivial partitions. So, when possible for a graph, we only considered partitions with at least two regions and required that each region contained at least one state that was not a boundary state. The description length of behavior (the "model evidence") was computed using R code supplied by Dr. Alec Solway.

**Tomov et al. (2020).** Tomov et al. (2020) [7] propose an account of task decomposition as inference over hierarchical structure. Their generative model of hierarchical structure assumes partitions are drawn from a Chinese Restaurant Process and additionally assumes that edges between states are more likely when the states are in the same region. Their model also

incorporates terms related to the task distribution and reward function that we expect to have minimal impact on our results because we do not vary the task distribution and the reward function for our task is a constant.

To incorporate this model, we used the analysis in the "Simulation Two: Bottleneck States" section in the publication [7]. The analysis models participants ($N = 40$) as sampling from a generative model over hierarchical graphs, then randomly sampling subgoals (three per participant) from the states connected to a "bridge" edge, which connects distinct regions in the hierarchical graph. Notably, pairs of regions are connected by a single "bridge" edge—this is subtly different from standard graph partitions, where different regions can be connected by any number of edges. All parameters were left as reported in the publication [7], with the exception of choice stochasticity $\epsilon$ which we made entirely deterministic by setting $\epsilon = 1.0$. To implement the published analyses, we used the publicly available code at https://github.com/tomov/chunking. We define the model prediction as the logarithm of the number of times a subgoal was sampled by this procedure—to avoid issues where a subgoal isn't sampled due to noise, we add 1 to the subgoal counts before taking the logarithm.

## Supporting information

**S1 Appendix. Appendix.** Extended data and analyses.
(PDF)

## Acknowledgments

We are very grateful to Dr. Alec Solway for providing software to compute the quantities defined in Solway et al. (2014) [8]. We thank Dr. Brendan McKay for making the eight-node graphs we analyze in this study available for download on his website. We thank Dr. Ari Kahn for helpful discussions about behavioral confounds.

## Author Contributions

**Conceptualization:** Carlos G. Correa, Mark K. Ho, Frederick Callaway, Nathaniel D. Daw, Thomas L. Griffiths.

**Data curation:** Carlos G. Correa.

**Formal analysis:** Carlos G. Correa, Mark K. Ho.

**Funding acquisition:** Nathaniel D. Daw, Thomas L. Griffiths.

**Investigation:** Carlos G. Correa.

**Methodology:** Carlos G. Correa, Mark K. Ho, Frederick Callaway.

**Project administration:** Carlos G. Correa.

**Software:** Carlos G. Correa, Mark K. Ho.

**Supervision:** Mark K. Ho, Nathaniel D. Daw, Thomas L. Griffiths.

**Validation:** Carlos G. Correa.

**Visualization:** Carlos G. Correa.

**Writing – original draft:** Carlos G. Correa, Mark K. Ho.

**Writing – review & editing:** Carlos G. Correa, Mark K. Ho, Frederick Callaway, Nathaniel D. Daw, Thomas L. Griffiths.

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
