## [Decision Letter · Decision Letter 0]

4 Jan 2023

Dear Mr. Correa,

Thank you very much for submitting your manuscript "Humans decompose tasks by trading off utility and computational cost" for consideration at PLOS Computational Biology. As with all papers reviewed by the journal, your manuscript was reviewed by members of the editorial board and by several independent reviewers. The reviewers appreciated the attention to an important topic. Based on the reviews, we are likely to accept this manuscript for publication, providing that you modify the manuscript according to the review recommendations.

We apologise for the delay in getting back to you - it has taken considerable time to solicit expert comments on this paper. As you can see from the review below, all reviewers liked the research question and appreciate the approach you have taken. They also liked the writing of the paper and we believe it will make a significant contribution to our journal. However, all reviewers had several comments and suggestions that we would like you to address in a revision of the paper.

Sincerely,

Tobias U Hauser, PhD

Academic Editor

PLOS Computational Biology

Thomas Serre

Section Editor

PLOS Computational Biology

Reviewer's Responses to Questions

**Comments to the Authors:**

Reviewer #1: The manuscript proposes a normative computational account of subgoal formation, which aims to integrate how the subgoals are chosen with the ways they are subsequently used, in the class of problems that translates into pathfinding problems on graphs. The authors implement a human behavioural experiment to assess the predictive power of the computational account and conclude that one of the graph search algorithms, one that closely corresponds to an interesting heuristic, corresponds to human choices of subgoals the best. In my opinion the paper is very thorough, thoughtful and very successful in integrating computational insight with empirical observation in a rather subtle domain. I think it would be a great addition to the scientific record - however, I believe addressing the following concerns would make it easier for readers to correctly contextualise the presented results:

There are three problem classes defined by the paper: the one described by the mathematical formalism of the model, the one implemented in the experiment, and the one from which the examples in the text are given. I think the text basically treats all these three to be the exact same, and I think they are not. I don’t necessarily think that this is a problem - abstracting away some aspects of the experimental setup and addressing a more general set of phenomena in the examples might even be desired - but I think writing very clearly about these differences is important to help the reader see what the results really mean. My first two questions address these differences.

1. The mathematical formalism assumes T to be known, and to be accessed locally (i.e. no searching from the goal backwards). This is fortunate since learning T would almost certainly be entangled with the choice of Z making the formalism way too complicated. However, in the experiment, people have to learn T from scratch, and aren’t expected to have a noiseless version of it at any point. In the navigation trials, this is compensated by the local information about edges that can be said to correspond to the local access in the model - but there people are making actual moves on the graph, not merely mental ones, as e.g. in the given example of chess. And in the probes, where they presumably are making mental simulations, they presumably not only learned T already (albeit noisily) but also already constructed Z - thus people aren’t really ‘planning’ while constructing the subgoals, but rather ‘exploring’. All this might be perfectly fine, it just gave me pause not to see these points addressed more directly.

2. The text of the manuscript gives various real-world examples of problems, most notably that of physical navigation, choosing landmarks (the café) as subgoals. I think the café example works through a different mechanism: I’d choose it as a subgoal because I already know how to get there, i.e. there is an existing (habitualised) policy I can reuse, and not because it will make planning cheaper now. Habits seem to be a consideration orthogonal to this paper, but maybe I’m failing to see a connection. Furthermore, in tasks like physical navigation people seem to heavily rely on distance-from-goal or directional heuristics, both available due to the existence of an embedding into a feature space. This could result in something like an IDA* search algorithm instead of the ones discussed here. No such embedding is assumed either in the model or the experiment - in fact it’s explicitly avoided. The example of navigating in the dark is closer to the experiment, however not the model. Board game examples are closer to the model, as they involve planning in symbolic spaces (but one has to ignore heuristic-generating feature embeddings for e.g. chess). Cooking might be the example I find the most fitting for the model.

3. The Discussion mentions in passing that this work is complementary to option discovery. This seems to be an important point to address, and although I myself am not well versed enough in the options literature to be able to tell the exact relationship, I’d be very interested how exactly is it complementary. In fact, I’d probably mention this in the introduction as well instead of only at the very end.

4. Relatedly, the learning problem in this paper is formalised similarly to goal-directed reinforcement learning. In particular, this paper looks at defining subgoals in such a setting: https://arxiv.org/abs/2106.01404. It might worth relating to this literature as well.

5. Why is IDDFS an intuitive choice despite the large computational cost depicted in Fig 1b? It has presumably been proposed previously because of some favorable property, was it good memory complexity? If so, would it also make sense to look at how subgoals reduce memory costs?

6. Do Fig 1c-e show the costs incurred during one particular run of each of these algorithms? Or why is e.g. the random walk not symmetric around the start state? These panels are also a bit far from the part of the text that describes them, moving them closer, maybe on a separate figure could streamline the reading experience.

7. In the experiment could stimulus salience distort the results in any way? E.g. is the red balloon overrepresented in the choice of subgoals?

8. Would it be possible to make more direct comparisons between the number of steps people make in the navigation trials and the number of steps taken by the algorithms? Would such a comparison be meaningful given that for humans this is a learning phase as well?

9. Eq 3: is the Z that maximises this formula taken here? I assume it is but it isn’t stated explicitly.

10. It is mentioned that the subjects were asked if they drew a map. Did any of them answer yes to this question?

In sum, I find the manuscript to be a valuable contribution, and regard the above issues as mere possible suboptimalities, not disqualifying problems. I look forward to seeing the final version of the paper, hopefully allowing me to understand the work even better.

Reviewer #2: The paper describes behavioral predictions derived from a set of (normative) computational accounts and heuristics for the (resource-rational) decomposition of tasks in a graph-structured environment. For simulations, graph structures are selected such that differences between model predictions about which states should be sub-goals become qualitatively evident. Model predictions are qualitatively and quantitatively (using multinomial regression analyses) compared against human behavior from a large, pre-registered online study (N = 806). The authors report that human behavior in a graph-structured planning task involving explicit and implicit probe questions about sub-goals is most consistent with the use of a task decomposition heuristic (betweenness centrality) and – among the formal accounts – with a resource-rational model performing an iterative-deepening depth-first search on the graph structure.

The paper is well-written, addresses an interesting (and novel) research question and features a variety of well-crafted computational accounts of task decomposition that are motivated by a resource-rational perspective on planning. Predictions of previously considered formal accounts of planning from the literature are pitted up against these new algorithms – allowing for quantitative comparisons of the relative goodness of fit to observed human behavior. The novelty and strength of the present computational approach lies in the formalization of three nested levels of planning (action-level planning, subtask-level planning and task decomposition), and their optimization considering computational costs (limited resources).

The idea that humans (and potentially other cognitive systems) engage in resource-rational trade-offs during planning and decomposition of tasks is intriguing, and has far-reaching implications, even beyond the field of cognitive psychology/neuroscience.

While I enjoyed reading the paper and think it would be of much interest to the diverse readership of PLOS Computational Biology, I have a few comments that I would like to see addressed. These are mainly related to a potential confound in the behavioral task design (that should at least be discussed), the presentation/analysis of the behavioral data and the interpretation of the findings. I am very confident that the authors will be able to address my concerns.

Major questions and comments:

1.) I would like to see the human behavior unpacked and explored a bit more:

a. A figure for the observed associations between navigation trial performance and probe behavior could be used to illustrate these findings. This would help readers to get a better sense for the variability of performance across subjects and the data distributions at hand.

b. In Figure 5a it is unclear how many participants performed choices on each of the depicted graphs. Please clarify this, and potentially consider adding a supplementary figure showing the results for the remaining 26 graph structures that were considered in the study.

c. A potential confound that should be controlled for is the varying complexity of the employed graph structures. Is discovery and usage of sub-goals further modulated by measures of graph complexity/minimum description length?

d. Does behavioral performance improve across trials/repeated exposures to planning tasks?

e. Is there evidence that participants learned the structure well-enough? What looks like absence of use of normative task decomposition could in fact be failure to acquire the structure. This learning deficit could be assessed by investigating exploration behavior before setting sub-goals (entropy in cursor/mouse movement, return to previously visited states – over and above the reported control for state occupancy during navigation trials) as marker of how well the structure has been learned.

f. Relatedly, was there evidence for (overall) longer reaction times on the task (e.g. longer planning duration, or time to complete a trial) for subjects choosing less optimal sub-goals (or no sub-goals at all), which would be expected if there are advantages of using a normative strategy to solve the task?

2.) Does task decomposition in the behavioral experiment occur “naturally”, or is it induced by task instructions, e.g. “Plan how to get from A to B. Choose a location you would visit along the way,” (lines 340-341), and subjects feel encouraged to do so; i.e. due to demand characteristics of the task? The authors should acknowledge and discuss this potential confound and its implications for the interpretation of the results. Additionally, it would be beneficial to point out that future studies should try to address this confound by using the same task but without explicitly prompting sub-goal use (and therefore task decomposition). This would allow to test whether the results are still aligned with the predictions of normative accounts and heuristics.

3.) It is unclear to me, how exactly the tested approximations/heuristics are more psychologically plausible and tractable for humans than the presented normative computations. As also discussed by the authors, betweenness centrality is computationally very demanding, given that all shortest paths of a given graph need to be computed and stored in memory to calculate the importance of each node using this heuristic. Relatedly, I would like to encourage the authors to include a section elaborating on how these heuristics (and potentially the computations used in the normative accounts) might plausibly be implemented by humans/brains (i.e. how cognitively and also how biologically plausible their implementation is). The authors acknowledge that even simpler heuristics than the ones considered here could be used by participants. I think it would be beneficial to elaborate more on what these simpler, more tractable heuristics may be. For example, simple count-based strategies, keeping track of the number of edges of and how often each node occurred in each query about start and end node seem to be a more tractable approximation (akin to something like a successor representation).

4.) The current study presents a normative account for resource-rational behavior in graph-structured environments, where sub-tasks are well-defined and the state space can be decomposed with reasonable certainty, only by considering the graph structure itself (subjects transition from one state to another, transitions do not depend on skill level of executing the behavioral task at hand, or the time to complete it etc). To keep up with the general motivation of the study as put forth in the author summary and introduction (“how do people decompose tasks to begin with?”, line 11), the authors could elaborate on the extent of how generalizable the presented normative accounts are to other, non-graph-structured planning tasks. How are sub-goals identified in the more general case, e.g. in more complex, less discrete tasks that involve more uncertainty about the state space and completion of sub-tasks/achievement of sub-goals?

5.) For resource-rational computations in subtask-level planning to occur, subjects would need to have access to the computational cost of a cognitive process – before deciding whether to engage in the computation, or rather not to invest time and cognitive resource. This cost is of course only available after indeed running the very same computation, which sort of seems to defeat the purpose of resource-rational deliberation. It is more of a question out of curiosity, but I assume many readers will have similar thoughts – so it might be beneficial to elaborate (e.g. in the discussion) how the necessary “ingredients” for the resource-rational deliberation are thought to be accrued before engaging in resource-rational task decomposition.

6.) It would be beneficial to present an additional, alternative metric for model comparison that is less dependent on the assumption of uninformative (flat) priors and an approximately multivariate Gaussian posterior distribution as the AIC. I suggest adding another metric like the WAIC, or preferably, cross-validation to assess the predictive accuracy of the models under consideration. Do other metrics produce convergent model comparison results?

Minor questions and comments

(1) It is surprising that predictions based on betweenness centrality seem so closely aligned with predictions of RRTD-IDDFS but not with RRTD-BFS (Fig. 3), given algorithmic work suggesting that betweenness centrality can be efficiently (and probabilistically) approximated using balanced bidirectional breadth-first search (e.g. Borassi & Natale, 2019, https://doi.org/10.1145%2F3284359). The authors could clarify and discuss this.

(2) The last sentence of the abstract “Taken together, our results provide new theoretical insight into the computational principles underlying the intelligent structuring of goal-directed behavior.”, seems to overstate the behavioral findings of the study and what the models represent. In my view, the study shows how well predictions of a number of considered normative accounts and heuristics are aligned with human behavior, but do not necessarily represent a proof for the use of these exact computational principles by humans (there could be alternative computational accounts and heuristics that are currently not considered in the model space of the present study – e.g. Dijkstra’s algorithm for discovery of the shortest path between nodes).

(3) I was a bit confused by the fact that the authors indicate the number of all possible unique 8-node graphs graph-structured planning tasks (11,117) multiple times throughout the manuscript without mentioning that this was not the number of graphs actually used in the behavioral experiment. It is more informative to learn that the authors further limited this set by ensuring that each graph had 10 distinct tasks with 3+ actions for an optimal solution (lines 532-533), which greatly enhances the scrutiny of the approach.

(4) In which way was the multiple-choice survey question at the end of the experiment used? Did it serve as an exclusion criterion? It would be beneficial to rule out the potentially confounding effects of participants using drawings or pictures of the graph and re-run the behavioral analyses only including subjects who did indeed adhere to the protocol.

(5) The explanations of the toy example task decomposition in the figure caption of Fig. 1 (c-e, page 3) are a bit unclear without reading the section describing how the task was set up in the main manuscript (only at page 6). Please expand the figure caption such that this becomes clearer without having to refer to the main text.

(6) Page 12, lines 354-355: I do not understand the pre-registered exclusion criterion of “no more than 175% of the optimal number of actions”. Is this a typo?

(7) Page 13, line 403: The last sentence before Figure 5 seems to overstate the behavioral findings. I do not think that the presented analyses (of internal consistency) are sufficient to establish validity of the construct (from a test theoretic perspective) – internal consistency is a metric of reliability. Please rephrase this.

(8) Figure 3: Why are there relatively low correlations between RRTD-RW and Q-Cut model predictions, if one is a heuristic approximation of a random walk, while correlations e.g. between RRTD-IDDFS and betweenness centrality are much higher. Was another than rank-one approximation used?

(9) The Github link to data and code used for analysis (https://github.com/cgc/resource-rational-task-decomposition) is currently not working, please make this important information available.

I sign my reviews.

Lennart Luettgau

Reviewer #3: In summary, my view is that this is a well-executed study which makes a significant contribution to the literature on human planning. In particular, the authors are to be commended on their efforts to integrate a variety of hitherto disparate studies within a unified perspective under the framework of resource-rational DM. Furthermore, a more detailed analysis of the hierarchy/bottleneck problem is presented based on the most comprehensive experiment on this topic to date. However, there are a couple of important gaps in the data analysis approach in my view.

Regarding the data analysis and model comparisons. The authors emphasize the algorithm-based approach in contrast to the structure inference approach. I agree with this perspective and find it interesting however it seems to me that this suggests an investigation into what planning algorithm is being used by the participants. To put it bluntly, what is the utility of considering a model such as RRTD-IDDFS to predict subgoals if the participants are not using IDDFS to plan? I wonder if the authors could at least provide some perspectives on this if not actually run some model fits/comparisons on choices during the navigation trials.

Related to this, it seems that an immediate computational hypothesis emerging from the normative framework studied here (and the principle of subgoaling more generally) regards the modulation of reaction times. That is, given a task decomposition, then subtask-level planning should occur at a subgoal specifically and this should be reflected in reaction times. More generally, reaction times can be an important behavioural indicator of internal computation and I think it should be somehow addressed in this study.

I think the authors could tune their introduction to the literature a bit better. For example, on the critical idea of relating task decompositions to planning (rather than structure inference), it is said that “…our framework differs from many existing accounts because we directly incorporate planning costs into the criteria used to choose a task decomposition.” I think it should be acknowledged that this idea is not fundamentally new and existing accounts have already considered planning costs in task decompositions computationally e.g. Jinnai et al 2019 (in RL) and McNamee et al 2016 (regarding human planning) (both cited here but there may be others). In particular, the latter considers a random walk search policy and points to log(degree centrality) as a key variable in determining decompositions/subgoals consistent with the modelling results here (see Fig 3 RRTD-RW vs degree centrality (log)). I think the specific computational novelty here is the integrative framework (which generates new results).

Minor comment:

Can authors speculate on the low consistency of teleportation probe? As I understand it, this measure is taken once per subject at the end of the experiment thus I would intuitively expect this measure of subgoals to be stable as opposed to the other measures which may be varying throughout the experiment.

**Have the authors made all data and (if applicable) computational code underlying the findings in their manuscript fully available?**

Reviewer #1: None

Reviewer #2: **No: **Link to Github repository was not working (as of 02/12/2022)

Reviewer #3: Yes

PLOS authors have the option to publish the peer review history of their article (what does this mean?). If published, this will include your full peer review and any attached files.

Reviewer #1: No

Reviewer #2: **Yes: **Lennart Luettgau

Reviewer #3: No

Figure Files:

Data Requirements:

Reproducibility:

References:

---

## [Decision Letter · Decision Letter 1]

10 Apr 2023

Dear Mr. Correa,

We are pleased to inform you that your manuscript 'Humans decompose tasks by trading off utility and computational cost' has been provisionally accepted for publication in PLOS Computational Biology. As you can see from the reviewers' comments, all of them were happy with the thorough revisions you have conducted and I congratulate you on this nice contribution to our field.

Best regards,

Tobias U Hauser, PhD

Academic Editor

PLOS Computational Biology

Thomas Serre

Section Editor

PLOS Computational Biology

Reviewer's Responses to Questions

**Comments to the Authors:**

Reviewer #1: The authors’ responses elaborate on each point I raised in my review of the initial submission, and the modifications of the text and the figures adequately address my questions and concerns. I find the revised manuscript to be significantly improved in terms of clarity and ease of understanding. Consequently, I see no further obstacles to the publication of this work.

Reviewer #2: The authors have addressed all of my concerns and questions through a convincing set of revisions to the manuscript, and by adding supplementary analyses and extending previous ones. I support publication of this highly intriguing and well-written paper and congratulate the authors on an excellent piece of work on resource-rational trade-offs during task decomposition and planning.

I sign my reviews.

Lennart Luettgau

Reviewer #3: Thank you for addressing my comments. I have no further comments and am happy to see this interesting study published.

**Have the authors made all data and (if applicable) computational code underlying the findings in their manuscript fully available?**

Reviewer #1: None

Reviewer #2: Yes

Reviewer #3: Yes

PLOS authors have the option to publish the peer review history of their article (what does this mean?). If published, this will include your full peer review and any attached files.

Reviewer #1: No

Reviewer #2: **Yes: **Lennart Luettgau

Reviewer #3: No

---

## [Editor Report · Acceptance letter]

8 May 2023

PCOMPBIOL-D-22-01548R1 

Humans decompose tasks by trading off utility and computational cost

Dear Dr Correa,

I am pleased to inform you that your manuscript has been formally accepted for publication in PLOS Computational Biology. Your manuscript is now with our production department and you will be notified of the publication date in due course.

With kind regards,

Zsofi Zombor
